# Cortical activity associated with the maintenance of balance during unstable stances

Shoma Ue, Kakeru Nakahama, Junpei Hayashi and Tomohiro Ohgomori

Department of Rehabilitation, Osaka Kawasaki Rehabilitation University, Kaizuka, Osaka, Japan

## ABSTRACT

**Background:** Humans continuously maintain and adjust posture during gait, standing, and sitting. The difficulty of postural control is reportedly increased during unstable stances, such as unipedal standing and with closed eyes. Although balance is slightly impaired in healthy young adults in such unstable stances, they rarely fall. The brain recognizes the change in sensory inputs and outputs motor commands to the musculoskeletal system. However, such changes in cortical activity associated with the maintenance of balance following periods of instability require further clarified.

**Methods:** In this study, a total of 15 male participants performed two postural control tasks and the center of pressure displacement and electroencephalogram were simultaneously measured. In addition, the correlation between amplitude of center of pressure displacement and power spectral density of electroencephalogram was analyzed.

**Results:** The movement of the center of pressure was larger in unipedal standing than in bipedal standing under both eye open and eye closed conditions. It was also larger under the eye closed condition compared with when the eyes were open in unipedal standing. The amplitude of high-frequency bandwidth (1–3 Hz) of the center of pressure displacement was larger during more difficult postural tasks than during easier ones, suggesting that the continuous maintenance of posture was required. The power spectral densities of the theta activity in the frontal area and the gamma activity in the parietal area were higher during more difficult postural tasks than during easier ones across two postural control tasks, and these correlate with the increase in amplitude of high-frequency bandwidth of the center of pressure displacement.

**Conclusions:** Taken together, specific activation patterns of the neocortex are suggested to be important for the postural maintenance during unstable stances.

Corresponding author
Tomohiro Ohgomori,
ohgomorit@kawasakigakuen.ac.jp

## INTRODUCTION

Humans continuously maintain and adjust posture during gait, standing, and sitting. Postural stability is sometimes evaluated using the movement activity of the center of pressure (COP). COP displacement is closely associated with fall risk, so the recording of

COP trajectory is widely used as a biomarker of fall risk in older people (*Quijoux et al., 2020*). Although postural stability is also impaired in healthy young adults by increase in postural task difficulty, they rarely fall (*Watanabe et al., 2018*; *Piitulainen et al., 2018*; *Wiesław Błaszczyk, Fredyk & Mikołaj Błaszczyk, 2020*). Healthy young adults are thus suggested to maintain their body balance continuously during unstable stances.

The motion of COP is composed of two components. Rambling reflects the motion of a reference point, and it is characterized by low-frequency bandwidths of COP movement (<0.5 Hz). On the other hand, trembling closely associated with the adjustments in balance, and it is characterized by middle to high-frequency fluctuations (0.5–2.0 Hz) of COP movement (*Zatsiorsky & Duarte, 1999*; *Yamagata, Popow & Latash, 2019*). Recent frequency analyses have shown that the power of the COP displacement at frequencies above 0.1 Hz is larger in unstable stances than in stable stances in healthy subjects (*Sim et al., 2018*). By contrast, the median-frequency of the COP displacement was shown to be decreased by lightly touching a finger on a table during eye closed (EC) condition, lead to the stabilization of posture (*Sozzi & Schieppati, 2022*). In addition, low-frequency bandwidth of the COP displacement is increased by aging (*Delmas et al., 2021*). Moreover, the COP displacement is decreased and mean frequency of COP displacement is contrary increased as the increase of postural threat in healthy adults (*Fischer et al., 2023*). We also reported that the fold change in amplitude of the COP displacement caused by the postural transition from bipedal (BP) to unipedal (UP) was significantly higher at high-frequency bandwidth than at low- and middle-frequency bandwidths (*Sugihara et al., 2021*). However, it is surprising that the high-frequency components of the COP displacement are also increased in patients with Parkinson's disease (PD) (*Matsuda et al., 2016*). Although the postural control mechanisms are not the same for PD patients, elderly, and healthy people performing UP standing under the EC condition, the large power for high-frequency components of the COP displacement is suggested by these reports to reflect the maintenance of balance due to elevated activity of calf muscles and resultant rigidity of the ankle joint, rather than a reflex or voluntary intervention from the central nervous system (*Zatsiorsky & Duarte, 1999*).

On the other hand, posture is maintained by not only a feedforward mechanism but also a feedback mechanism based on visual, auditory, and proprioceptive sensory information (*Prosperini & Castelli, 2018*). Cortical activity was shown to be significantly altered in continuous postural tasks of increasing difficulty due to changes in afferent inputs (*Edwards et al., 2018*). Recently, we and other groups reported that the prefrontal cortex was activated during difficult postural tasks in healthy young adults (*St George et al., 2021*; *Sugihara et al., 2021*). Although there are many previous reports about the change in cortical activity during postural control tasks, the change in cortical activity closely related to a feedback mechanism of postural control has not been fully clarified. We considered that cortical activities related to a feedback mechanism rather than a feedforward one may be identified by the investigation of relationship between the power for the high-frequency components of the COP displacement and the change in cortical activities.

Electroencephalography is a representative neuroimaging tool that can wirelessly measure the physiological activity of a wide range of cortical areas, and is not significantly

affected by the presence or absence of hair. Five major frequency bandwidths of electroencephalogram (EEG) are known: delta (2–4 Hz), theta (4–8 Hz), alpha (8–13 Hz), beta (13–30 Hz), and gamma (more than 30 Hz) (*Campisi & La Rocca, 2014*; *Kiiski et al., 2020*; *Chen et al., 2023*). The effects of postural control task difficulty on EEG are well studied. For instance, frontal theta band activity was increased and parietal alpha band activity was decreased during difficult postural control tasks in healthy subjects (*Edwards et al., 2018*; *Gebel, Lehmann & Granacher, 2020*; *Malcolm et al., 2021*). The reduction of alpha power is also observed in transfemoral amputees that postural adjustment is difficult (*Khajuria & Joshi, 2022*). Beta rhythm (13–17 Hz) in the supplementary motor area was suppressed by dynamic movements, such as stepping and feet-in-place (*Solis-Escalante et al., 2019*). Moreover, unpredictable perturbations of upright stability induce characteristic event-related potentials in the fronto-central region and the reorganization of functional networks in fronto-centro-parietal area of the cerebral cortex (*Varghese et al., 2014*; *Varghese, Staines & McIlroy, 2019*). There are many previous studies evaluating the cortical activities related to postural control using easy and difficult balance tasks. However, few studies have shown the cortical activities commonly observed during several postural control tasks, whose difficulty was increased in different ways, comparatively. In addition, EEG data measured during individual postural control tasks may contain many changes in EEG that are mainly related to sensory alterations independent for the postural maintenance, such as the deprivation of visual inputs. To address these issues, it is necessary to clarify the common changes in EEG related to the postural maintenance using several postural control tasks which difficulties are increased in different ways. It is possible to exclude cortical activities independent of postural maintenance using this strategy.

In this study, we hypothesized that sensory feedback might induce specific common changes in cortical activity during the maintenance of posture under various unstable stances. To test this hypothesis, we simultaneously measured EEG and the COP displacement during two types of postural control task from easy to difficult in a cohort of healthy young adults based on previous studies. One is the change in stances from BP to UP, and the other is eye closure. In addition, we examined the correlation between the change in power spectral density of EEG and the amplitude of high-frequency bandwidth of the COP displacement to investigate the change in cortical activity related to a feedback mechanism of postural control.

# MATERIALS AND METHODS

## Participants

In this study, 15 young male adults (age = 21.2 ± 0.41 years old, height = 170.6 ± 4.44 cm, body weight = 70.0 ± 11.7 kg) were recruited from college students. All participants were volunteers and their ethnicity was Japanese, and they were not minors. Sample size in present pilot study was determined using G*Power Ver 3.1.9.7 according to a previous report (15 subjects, medium effect size d = 0.8, $\alpha$ = 0.05, statistical power (1-$\beta$ error probability) = 0.8, statistical test type = matched pairs) (*Cohen, 1992*). The research subjects were all healthy and had no orthopedic, neurological, or ophthalmic diseases. Exclusion criteria were set as follows. Since aging significantly affect to the COP

movement, children (<15 years), and senior (>50 years) were excluded (*Kurz et al., 2018*). Since sex differences are also known to affect postural sway characteristics in young participants, male adults were included and female adults were excluded in this study (*Kozinc et al., 2021*; *Jo et al., 2022*). Young adults with a history of epilepsy were excluded because it was known that abnormal EEG patterns were detected. In addition, patients with the history of injury in anterior cruciate ligament and lower limb musculoskeletal systems were excluded according to a previous review (*Lehmann, Paschen & Baumeister, 2017*). Other exclusion criteria based on physical functions (*e.g.*, muscle strength and endurance, joint range of motion, and habits of exercise) were not set. Visual acuity was separately tested in each eye using Landolt C chart in a random order. The participants who have a visual acuity of <1.0, with their glasses on or with the naked eye, were excluded according to the previous our study (*Nakahara et al., 2022*). The dominant foot of all participants was the right foot. All experiments were conducted in accordance with the Code of Ethics of the World Medical Association (Declaration of Helsinki) and were approved by the Ethics Committee of Osaka Kawasaki Rehabilitation University. Osaka Kawasaki Rehabilitation University granted Ethical approval to carry out this study within its facilities (Ethical Application Reference number: OKRU-RA0023). Written informed consent was obtained from all participants for publication of this study. The start and end of the recruitment period for this study was August 26, 2022 and March 27, 2023, respectively. To avoid assentation, all results were not communicated to the participants until the completion of the measurement schedule.

## Postural control task 1

The EEG measurement and postural control task were performed in an electrically-shielded, sound-attenuated, and shaded (<5 lux) space. Since balance impairments are often observed immediately after the postural and sensory changes such as BP-to-UP standing, sit-to-stand movements, and eye closure, we continuously performed easy and difficult postural control tasks in the same session in this study. Postural control task 1 from BP to UP standing was performed similarly to a previous report, with minor modifications (*Sabashi et al., 2021*). A 27-inch monitor (width = 59.66 cm, height = 33.60 cm) was set 60 cm in front of the stabilometer that is an instrument to measure center of gravity trajectory (T.K.K.5810; Takei Scientific Instruments Co., Ltd, Niigata, Japan). It was placed slightly below eye level to avoid contamination of EEG by artifacts derived from frontal muscle activity. Participants viewed a static image presented on the monitor. During BP standing, they stood at the center of the stabilometer with their heels aligned and their toes pointing forward. Load cells (PDA-3MPB; Tokyo Measuring Instruments Laboratory Co., Ltd, Tokyo, Japan) were attached to the position where the heels touched the stabilometer. Pressure on load cells was automatically recorded using a multi-recorder control unit (TMR-211; Tokyo Measuring Instruments Laboratory Co., Ltd, Tokyo, Japan). The distance between the left and right feet matched the participant's shoulder width. Under the eye open (EO) condition, the displacement of the COP was measured during BP standing for 30 s. Subsequently, the standing pattern was quickly changed to UP, and displacement of the COP was measured for 30 s. In the UP standing position, the

participants placed their nondominant foot as support (left, based on their self-reported kicking preference) on the stabilometer, and the hip and knee of the lifted right leg were flexed at 45° (*Promsri, Haid & Federolf, 2018*). The same postural control task was conducted under the EC condition. The interval of sessions was set at 2 min. Sound indications were provided to the participants when they changed their standing patterns. We verified that the participants lifted the sole of their dominant foot from the stabilometer, and the load on the heel of dominant foot was zero during UP standing. The sampling rate was 100 Hz. Measurements under the EO and EC conditions were performed twice each, and the order of trials was randomly presented to participants. To minimize the effect of fatigue and practice due to the repetition of multiple postural control tasks on the COP trajectories, the number of testing trials was set to two for each condition. The average of each trial was used for further analyses. The participants were instructed to cross their arms in front of their bodies to avoid using their upper limbs to balance, and to suppress body sway as much as possible during balance tasks. Since the unstable posture was controlled following the sound indication, the periods before and after the sound indication (30 s stance periods) were analyzed as the BP and UP phases, respectively.

### Postural control task 2

The stabilometer and load cells were set same as postural control task 1. The recording of displacement of COP and pressure on load cells were also same as postural control task 1. In BP standing, participants stood at the center of the stabilometer with their heels aligned and their toes pointing forward under the EO condition. The distance between the left and right feet again matched the shoulder width. The displacement of the COP was measured under the EO condition for 30 s. Subsequently, the visual inputs were deprived by the closure of their eyes, and the displacement of the COP was measured for 30 s. The same postural control task was conducted during UP standing. The interval between sessions was set at 2 min. In UP standing, the participants placed the nondominant foot, and the position of hip and knee of the lifted leg were same as postural control task 1. Sound indications were provided to the participants when they closed their eyes. We verified that the participants lifted the sole of their dominant foot from the stabilometer, and the load on the heel of dominant foot was zero during UP standing. Measurements in BP and UP standing were performed twice each, and the order of trials was randomly presented to participants. The average of each trial was used for further analyses. The behavioral instruction to participants during the balance task were the same as those for postural control task 1. Since the unstable posture was controlled following the sound indication, the periods before and after the sound indication (30 s stance periods) were analyzed as the EO and EC phases, respectively.

### Loading data processing

To verify the loading pressure on the sole during UP standing and the time point of change in loading pressure, a four-second segment (29–33 s) was extracted from the time-series data of loading pressure on dominant foot. Extracted time-series data was approximated

with several standard linear regression models using a strucchange package in R, and the intersection points of these models were defined as breaking points (BKs) (Zeileis et al., 2002). We determined the number of standard linear regression models, by which the Bayesian information criterion (BIC) became the minimum. BIC values were automatically calculated based on the following formula.

$$\mathrm{BIC} = -2\ln(L) + k\ln(n)$$

$L$: the maximized likelihood of the selected model
$k$: the corresponding number of parameters
$n$: the sample size

Two BKs were identified in all time-series data. The difference between the individual values and the median of BKs were examined.

## Eye tracking

Eye tracking was used for the determination of eye close time point. Eye tracking was performed using Tobii eye tracker 5 (Tobii Technology K.K., Stockholm, Sweden) according to a previously reported method (Nakahara et al., 2022). The gaze point was changed to the cursor position using Miyasuku EyeConLT2 (Unicorn Corp., Hiroshima, Japan), and the coordinates of cursor position were recorded by Python (JetBrains Self-regulatory Organization, Prague, Czech Republic). Pupil center corneal reflection algorithm was used for the tracking of eye position, so the motion of the cursor was completely stopped by eye closure (US patent us7572008; publication date 2009-08-11).

## Electroencephalography

Active electrodes were placed at Fp1, Fp2, F3, F4, C3, C4, P3 and P4 positions based on the international 10–20 system using an EEG cap with 32 electrodes. In this study, we selected frontal, central, and parietal areas, which are involved in motor regulation and sensory information processing (Oostra et al., 2016; Iandolo et al., 2018; Zhao et al., 2019). Participants' head circumference was measured to select the optimal BrainCap size (inner circumference = 54 and 56 cm; Brain Products GmbH, Gilching, Germany). Ground reference and reference electrodes were placed at Fpz and FCz positions, respectively. Based on a previous report, the electrode impedances were adjusted up to 5 kΩ and the value of electrode impedance was confirmed before the starting sessions (Górecka & Makiewicz, 2019). EEG signals were recorded at 1,000 Hz using an 8-channnel LiveAmp wireless mobile amplifier (Brain Products GmbH, Gilching, Germany), and were acquired using BrainVision Recorder (Brain Products GmbH, Gilching, Germany). A trigger pulse (+5V, 100 ms) was furnished to the wireless mobile amplifier, the stabilometer, and the multi-recorder control unit using a switch box (Takei Scientific Instruments Co., Ltd, Tokyo, Japan) to synchronize time-series data. Participants were instructed not to swallow and bite during the session to avoid contamination of artifacts derived from muscle activity.

## EEG data processing

EEG data processing was performed using BrainVision Analyzer 2 (Brain Products GmbH, Gilching, Germany). To remove body movement noise, the EEG signals were filtered by infinite impulse response bandpass (2–100 Hz, second-order Butterworth) filter. Notch filter was applied to remove 60 Hz electronical power line noise. Then, the artifacts derived from eye blink, which showed strong topography in Fp1 and Fp2 channels, were automatically identified using independent component analysis (ICA) (*Edwards et al., 2018*). Since sphering is generally considered a necessary precondition to ICA, the classic sphering method was used in this study. The sphered data was fed into Restricted Fast ICA to find an optimal unmixing matrix. (*Hyvärinen & Oja, 2000*). After the filtering, we visually confirmed that the EEG signals did not contain abnormal potential drift noises according to set criteria described previously with minor modification: maximal allowed voltage step of 75 μV/ms; maximal allowed absolute difference of 250 μV (interval length 400 ms); maximal/minimal allowed amplitude of ±175 μV; and lowest allowed activity of 0.5 μV (interval length 100 ms) (*Jansen et al., 2020*; *Rosjat et al., 2021*). Prior to spectral analysis, EEG signals for 60 s after the trigger pulse were segmented. These were subsequently segmented into 30 s epochs before and after the increase in postural control difficulty. Additionally, the EEG data of 2 s before and after the sound indication including non-stationary responses was eliminated, and the remaining EEG data of 28 s were applied to following spectral analysis. Power spectral densities (PSDs) were calculated by fast Fourier transformation (FFT) according to set criteria: Hanning window; windows length 10% with variance correction; periodic windows function; resolution 0.031 Hz.
The following formula was used for the calculation of PSD in each frequency.

$$\mathrm{PSD} = \mathrm{Amplitude}^2/(\Delta \mathrm{f} \times \mathrm{Wf})$$

Wf represents the correction value for each window function, Hanning = 3/2

Δf represents the frequency resolution

The average of PSD of EEG in each frequency before the increase in postural control difficulty was used as the baseline, and the relative change in PSD of EEG after the increase in postural control difficulty was calculated.

## COP data processing

Based on results of loading data processing, the COP data of 2 s before and after the sound indication including non-stationary responses was eliminated. Prior to FFT analysis, we examined the distributions of standard deviation (SD) of the COP displacement in the anteroposterior (AP) and mediolateral (ML) directions. There were no clear data outliers in this study. The path length was calculated as the sum of the distances between the position of the COP point. The amplitude of the COP displacement was calculated using the temporal coordinate data of COP in AP and ML directions. All temporal coordinates of COP in AP and ML directions were transformed to frequencies using Bluestein's FFT according to a previous study (*Sugihara et al., 2021*). These signals were low pass filtered with a cut-off at 3.0 Hz based using a second order Butterworth filter, as used in a previous report (*Loram, Gawthrop & Lakie, 2006*). The power spectrum was divided into three

**Table 1 Statistical summary of the two-way ANOVA in the analysis of COP displacement.**

| | Stance × Eye | | Stance | | | | Eye | | | |
|---|---|---|---|---|---|---|---|---|---|---|
| | | | EC | | EO | | UP | | BP | |
| Figure no. | *F*-value | *P*-value | *F*-value | *P*-value | *F*-value | *P*-value | *F*-value | *P*-value | *F*-value | *P*-value |
| 3C | $F_{1,14}$ = 130.20 | <0.001 | $F_{1,14}$ = 145.20 | <0.001 | $F_{1,14}$ = 23.12 | <0.001 | $F_{1,14}$ = 189.72 | <0.001 | $F_{1,14}$ = 138.72 | <0.001 |
| 3D | $F_{1,14}$ = 103.96 | <0.001 | $F_{1,14}$ = 129.82 | <0.001 | $F_{1,14}$ = 165.71 | <0.001 | $F_{1,14}$ = 38.66 | <0.001 | $F_{1,14}$ = 123.12 | <0.001 |
| 3E | $F_{1,14}$ = 144.10 | <0.001 | $F_{1,14}$ = 119.09 | <0.001 | $F_{1,14}$ = 190.97 | <0.001 | $F_{1,14}$ = 6.84 | 0.02 | $F_{1,14}$ = 141.99 | <0.001 |
| 3F | $F_{1,14}$ = 32.87 | <0.001 | $F_{1,14}$ = 112.84 | <0.001 | $F_{1,14}$ = 97.62 | <0.001 | $F_{1,14}$ = 7.65 | 0.015 | $F_{1,14}$ = 57.61 | <0.001 |
| 3G | $F_{1,14}$ = 134.50 | <0.001 | $F_{1,14}$ = 96.58 | <0.001 | $F_{1,14}$ = 177.98 | <0.001 | $F_{1,14}$ = 3.08 | 0.101 | $F_{1,14}$ = 123.05 | <0.001 |
| 4C | $F_{1,14}$ = 88.47 | <0.001 | $F_{1,14}$ = 166.09 | <0.001 | $F_{1,14}$ = 176.98 | <0.001 | $F_{1,14}$ = 45.66 | <0.001 | $F_{1,14}$ = 96.10 | <0.001 |
| 4D | $F_{1,14}$ = 51.84 | <0.001 | $F_{1,14}$ = 134.38 | <0.001 | $F_{1,14}$ = 102.77 | <0.001 | $F_{1,14}$ = 43.10 | <0.001 | $F_{1,14}$ = 58.39 | <0.001 |
| 4E | $F_{1,14}$ = 88.28 | <0.001 | $F_{1,14}$ = 141.77 | <0.001 | $F_{1,14}$ = 192.28 | <0.001 | $F_{1,14}$ = 13.81 | 0.002 | $F_{1,14}$ = 92.88 | <0.001 |
| 4F | $F_{1,14}$ = 19.13 | <0.001 | $F_{1,14}$ = 47.93 | <0.001 | $F_{1,14}$ = 31.30 | <0.001 | $F_{1,14}$ = 0.057 | 0.815 | $F_{1,14}$ = 17.46 | <0.001 |
| 4G | $F_{1,14}$ = 85.01 | <0.001 | $F_{1,14}$ = 117.32 | <0.001 | $F_{1,14}$ = 194.74 | <0.001 | $F_{1,14}$ = 4.16 | 0.061 | $F_{1,14}$ = 89.44 | <0.001 |

bandwidths: low (0–0.3 Hz), middle (0.3–1.0 Hz), and high (1.0–3.0 Hz) (*Nagy et al., 2004*; *Vieira et al., 2015*; *Nagymáté & Kiss, 2016*). The area under the spectral plots of amplitude in each frequency bandwidth was then calculated.

## Statistical analysis

Before the statistical analysis, normality was checked using the Kolmogorov-Smirnov test using R software. The null hypothesis is that data follows a normal distribution. The null hypothesis was not rejected. Data was statistically analyzed using KaleidaGraph 4.5 (Hulinks, Tokyo, Japan). The effect of eye closure on the loading pressure and the effect of standing pattern on eye closure were statistically analyzed by paired *t* test. In addition, the effect of standing pattern or eye closure on path length, amplitude of the COP displacement, and PSD of EEG were also statistically analyzed. The statistical differences between multiple groups were analyzed by repeated two-way analysis of variance (ANOVA) using the statistical analysis and the data mining software JASP (https://jasp-stats.org/). When there were significant interactions between two factors (stance and visual input), the analysis of simple main effects was further performed for both factors. The summary of repeated two-way ANOVA was shown in Tables 1–5. A *P*-value < 0.05 was considered statistically significant. For simplicity, we described only the significant effect of stance and eye closure.

To reveal common cortical activities related to postural maintenance under unstable conditions, we focused on postural tasks in which the amplitude of high-frequency bandwidth was increased. Then, the correlation between the change in amplitude of high-frequency bandwidth and the change in PSD of EEG was analyzed by following method. We first calculated change in PSD of EEG (dx) and change in postural sway amplitude of high-frequency bandwidth (dy) before and after the increase in postural

**Table 2** Statistical summary of the two-way ANOVA in case of non-significant interaction in main figures.

| Figure no. | Stance × Eye F-value | Stance × Eye P-value | Stance F-value | Stance P-value | Eye F-value | Eye P-value |
|---|---|---|---|---|---|---|
| $5B_1$ | $F_{1,14} = 0.417$ | 0.529 | $F_{1,14} = 1.27$ | 0.279 | $F_{1,14} = 5.14$ | 0.04 |
| $5B_2$ | $F_{1,14} = 5.84 \times 10^{-4}$ | 0.981 | $F_{1,14} = 0.561$ | 0.466 | $F_{1,14} = 7.12$ | 0.018 |
| $5C_1$ | $F_{1,14} = 0.218$ | 0.648 | $F_{1,14} = 0.182$ | 0.676 | $F_{1,14} = 1.90$ | 0.19 |
| $5C_2$ | $F_{1,14} = 1.95$ | 0.184 | $F_{1,14} = 4.09$ | 0.063 | $F_{1,14} = 14.09$ | 0.002 |
| $5D_1$ | $F_{1,14} = 0.257$ | 0.62 | $F_{1,14} = 7.99$ | 0.013 | $F_{1,14} = 3.74$ | 0.074 |
| $6A_1$ | $F_{1,14} = 2.20$ | 0.16 | $F_{1,14} = 10.95$ | 0.005 | $F_{1,14} = 27.66$ | <0.001 |
| $6A_2$ | $F_{1,14} = 0.789$ | 0.389 | $F_{1,14} = 1.44$ | 0.25 | $F_{1,14} = 3.98$ | 0.066 |
| $6B_1$ | $F_{1,14} = 0.077$ | 0.786 | $F_{1,14} = 1.07$ | 0.319 | $F_{1,14} = 6.91$ | 0.02 |
| $6B_2$ | $F_{1,14} = 0.851$ | 0.372 | $F_{1,14} = 0.543$ | 0.473 | $F_{1,14} = 6.70$ | 0.021 |
| $6C_1$ | $F_{1,14} = 1.69$ | 0.215 | $F_{1,14} = 0.112$ | 0.743 | $F_{1,14} = 0.013$ | 0.91 |
| $6C_2$ | $F_{1,14} = 0.378$ | 0.549 | $F_{1,14} = 0.231$ | 0.638 | $F_{1,14} = 11.71$ | 0.004 |
| $6D_1$ | $F_{1,14} = 0.607$ | 0.449 | $F_{1,14} = 2.41$ | 0.143 | $F_{1,14} = 0.986$ | 0.338 |

**Table 3** Statistical summary of the two-way ANOVA in case of non-significant interaction in Fig. S1.

| Figure no. | Stance × Eye F-value | Stance × Eye P-value | Stance F-value | Stance P-value | Eye F-value | Eye P-value |
|---|---|---|---|---|---|---|
| $S1A_1$ | $F_{1,14} = 4.19$ | 0.06 | $F_{1,14} = 25.73$ | <0.001 | $F_{1,14} = 4.05$ | 0.064 |
| $S1A_3$ | $F_{1,14} = 1.63$ | 0.223 | $F_{1,14} = 30.01$ | <0.001 | $F_{1,14} = 1.99$ | 0.181 |
| $S1A_4$ | $F_{1,14} = 0.056$ | 0.816 | $F_{1,14} = 11.93$ | 0.004 | $F_{1,14} = 11.17$ | 0.005 |
| $S1A_5$ | $F_{1,14} = 0.081$ | 0.78 | $F_{1,14} = 9.84$ | 0.007 | $F_{1,14} = 11.75$ | 0.004 |
| $S1A_6$ | $F_{1,14} = 4.56$ | 0.051 | $F_{1,14} = 1.53$ | 0.237 | $F_{1,14} = 5.56$ | 0.033 |
| $S1B_1$ | $F_{1,14} = 0.741$ | 0.404 | $F_{1,14} = 1.03$ | 0.328 | $F_{1,14} = 4.27$ | 0.058 |
| $S1B_2$ | $F_{1,14} = 0.581$ | 0.458 | $F_{1,14} = 1.10$ | 0.313 | $F_{1,14} = 4.05$ | 0.064 |
| $S1B_3$ | $F_{1,14} = 0.173$ | 0.683 | $F_{1,14} = 1.00$ | 0.334 | $F_{1,14} = 6.27$ | 0.025 |
| $S1B_4$ | $F_{1,14} = 4.17$ | 0.061 | $F_{1,14} = 4.98$ | 0.043 | $F_{1,14} = 9.78$ | 0.007 |
| $S1B_5$ | $F_{1,14} = 0.06$ | 0.810 | $F_{1,14} = 3.64$ | 0.077 | $F_{1,14} = 9.81$ | 0.007 |
| $S1B_6$ | $F_{1,14} = 1.03$ | 0.328 | $F_{1,14} = 1.93$ | 0.187 | $F_{1,14} = 6.98$ | 0.019 |
| $S1C_1$ | $F_{1,14} = 0.634$ | 0.439 | $F_{1,14} = 1.27$ | 0.279 | $F_{1,14} = 3.51$ | 0.082 |
| $S1C_2$ | $F_{1,14} = 0.11$ | 0.745 | $F_{1,14} = 4.87 \times 10^{-4}$ | 0.983 | $F_{1,14} = 2.79$ | 0.117 |
| $S1C_3$ | $F_{1,14} = 0.943$ | 0.348 | $F_{1,14} = 1.57$ | 0.231 | $F_{1,14} = 2.16$ | 0.164 |
| $S1C_4$ | $F_{1,14} = 2.24$ | 0.157 | $F_{1,14} = 1.03$ | 0.328 | $F_{1,14} = 27.18$ | <0.001 |
| $S1C_5$ | $F_{1,14} = 3.00$ | 0.105 | $F_{1,14} = 3.11$ | 0.1 | $F_{1,14} = 15.43$ | 0.002 |
| $S1C_6$ | $F_{1,14} = 1.62$ | 0.223 | $F_{1,14} = 2.39$ | 0.144 | $F_{1,14} = 19.06$ | <0.001 |
| $S1D_1$ | $F_{1,14} = 0.249$ | 0.626 | $F_{1,14} = 9.65$ | 0.008 | $F_{1,14} = 1.91$ | 0.189 |
| $S1D_2$ | $F_{1,14} = 0.865$ | 0.368 | $F_{1,14} = 2.43$ | 0.141 | $F_{1,14} = 0.326$ | 0.577 |
| $S1D_3$ | $F_{1,14} = 3.18$ | 0.095 | $F_{1,14} = 8.60$ | 0.011 | $F_{1,14} = 0.028$ | 0.87 |
| $S1D_4$ | $F_{1,14} = 0.002$ | 0.963 | $F_{1,14} = 2.00$ | 0.179 | $F_{1,14} = 0.035$ | 0.854 |
| $S1D_5$ | $F_{1,14} = 1.06$ | 0.321 | $F_{1,14} = 7.01$ | 0.019 | $F_{1,14} = 0.688$ | 0.421 |

**Table 4** Statistical summary of the two-way ANOVA in case of non-significant interaction in Fig. S2.

| | Stance × Eye | | Stance | | Eye | |
|---|---|---|---|---|---|---|
| Figure no. | *F*-value | *P*-value | *F*-value | *P*-value | *F*-value | *P*-value |
| $S2A_1$ | $F_{1,14}$ = 1.84 | 0.197 | $F_{1,14}$ = 5.87 | 0.03 | $F_{1,14}$ = 15.10 | 0.002 |
| $S2A_2$ | $F_{1,14}$ = 3.97 | 0.066 | $F_{1,14}$ = 7.31 | 0.017 | $F_{1,14}$ = 19.49 | <0.001 |
| $S2A_3$ | $F_{1,14}$ = 1.35 | 0.266 | $F_{1,14}$ = 4.19 | 0.06 | $F_{1,14}$ = 23.31 | <0.001 |
| $S2A_4$ | $F_{1,14}$ = 0.005 | 0.946 | $F_{1,14}$ = 3.43 | 0.085 | $F_{1,14}$ = 21.49 | <0.001 |
| $S2A_5$ | $F_{1,14}$ = 0.006 | 0.941 | $F_{1,14}$ = 2.98 | 0.106 | $F_{1,14}$ = 11.74 | 0.004 |
| $S2A_6$ | $F_{1,14}$ = 0.694 | 0.419 | $F_{1,14}$ = 1.03 | 0.328 | $F_{1,14}$ = 4.24 | 0.059 |
| $S2B_1$ | $F_{1,14}$ = 0.210 | 0.654 | $F_{1,14}$ = 0.271 | 0.611 | $F_{1,14}$ = 4.07 | 0.063 |
| $S2B_2$ | $F_{1,14}$ = 0.369 | 0.553 | $F_{1,14}$ = 0.802 | 0.386 | $F_{1,14}$ = 3.51 | 0.082 |
| $S2B_3$ | $F_{1,14}$ = 0.038 | 0.849 | $F_{1,14}$ = 1.05 | 0.323 | $F_{1,14}$ = 6.81 | 0.021 |
| $S2B_4$ | $F_{1,14}$ = 0.381 | 0.547 | $F_{1,14}$ = 3.33 | 0.089 | $F_{1,14}$ = 8.68 | 0.011 |
| $S2B_5$ | $F_{1,14}$ = 0.205 | 0.658 | $F_{1,14}$ = 1.27 | 0.279 | $F_{1,14}$ = 10.24 | 0.006 |
| $S2B_6$ | $F_{1,14}$ = 1.47 | 0.246 | $F_{1,14}$ = 1.19 | 0.294 | $F_{1,14}$ = 5.72 | 0.031 |
| $S2C_1$ | $F_{1,14}$ = 0.138 | 0.716 | $F_{1,14}$ = 2.40 | 0.144 | $F_{1,14}$ = 5.94 | 0.029 |
| $S2C_2$ | $F_{1,14}$ = 0.008 | 0.932 | $F_{1,14}$ = 0.01 | 0.923 | $F_{1,14}$ = 0.224 | 0.643 |
| $S2C_3$ | $F_{1,14}$ = 1.76 | 0.206 | $F_{1,14}$ = 0.022 | 0.884 | $F_{1,14}$ = 11.88 | 0.004 |
| $S2C_4$ | $F_{1,14}$ = 0.795 | 0.388 | $F_{1,14}$ = 0.324 | 0.578 | $F_{1,14}$ = 10.13 | 0.007 |
| $S2C_5$ | $F_{1,14}$ = 0.375 | 0.55 | $F_{1,14}$ = 1.13 | 0.307 | $F_{1,14}$ = 12.55 | 0.003 |
| $S2C_6$ | $F_{1,14}$ = 0.229 | 0.639 | $F_{1,14}$ = 0.162 | 0.693 | $F_{1,14}$ = 16.45 | 0.001 |
| $S2D_1$ | $F_{1,14}$ = 0.464 | 0.507 | $F_{1,14}$ = 6.69 | 0.022 | $F_{1,14}$ = 0.474 | 0.503 |
| $S2D_2$ | $F_{1,14}$ = 0.225 | 0.643 | $F_{1,14}$ = 2.98 | 0.106 | $F_{1,14}$ = 0.448 | 0.514 |
| $S2D_3$ | $F_{1,14}$ = 2.38 | 0.146 | $F_{1,14}$ = 3.24 | 0.094 | $F_{1,14}$ = 2.18 | 0.162 |
| $S2D_4$ | $F_{1,14}$ = 2.75 | 0.12 | $F_{1,14}$ = 0.088 | 0.771 | $F_{1,14}$ = 1.84 | 0.197 |

**Table 5** Statistical summary of the two-way ANOVA in case of significant interaction.

| | Stance × Eye | | Stance | | | | Eye | | | |
|---|---|---|---|---|---|---|---|---|---|---|
| | | | EC | | EO | | UP | | BP | |
| Figure no. | *F*-value | *P*-value | *F*-value | *P*-value | *F*-value | *P*-value | *F*-value | *P*-value | *F*-value | *P*-value |
| 3C | $F_{1,14}$ = 130.20 | <0.001 | $F_{1,14}$ = 145.20 | <0.001 | $F_{1,14}$ = 23.12 | <0.001 | $F_{1,14}$ = 189.72 | <0.001 | $F_{1,14}$ = 138.72 | <0.001 |
| 3D | $F_{1,14}$ = 103.96 | <0.001 | $F_{1,14}$ = 129.82 | <0.001 | $F_{1,14}$ = 165.71 | <0.001 | $F_{1,14}$ = 38.66 | <0.001 | $F_{1,14}$ = 123.12 | <0.001 |
| 3E | $F_{1,14}$ = 144.10 | <0.001 | $F_{1,14}$ = 119.09 | <0.001 | $F_{1,14}$ = 190.97 | <0.001 | $F_{1,14}$ = 6.84 | 0.02 | $F_{1,14}$ = 141.99 | <0.001 |
| 3F | $F_{1,14}$ = 32.87 | <0.001 | $F_{1,14}$ = 112.84 | <0.001 | $F_{1,14}$ = 97.62 | <0.001 | $F_{1,14}$ = 7.65 | 0.015 | $F_{1,14}$ = 57.61 | <0.001 |
| 3G | $F_{1,14}$ = 134.50 | <0.001 | $F_{1,14}$ = 96.58 | <0.001 | $F_{1,14}$ = 177.98 | <0.001 | $F_{1,14}$ = 3.08 | 0.101 | $F_{1,14}$ = 123.05 | <0.001 |
| 4C | $F_{1,14}$ = 88.47 | <0.001 | $F_{1,14}$ = 166.09 | <0.001 | $F_{1,14}$ = 176.98 | <0.001 | $F_{1,14}$ = 45.66 | <0.001 | $F_{1,14}$ = 96.10 | <0.001 |
| 4D | $F_{1,14}$ = 51.84 | <0.001 | $F_{1,14}$ = 134.38 | <0.001 | $F_{1,14}$ = 102.77 | <0.001 | $F_{1,14}$ = 43.10 | <0.001 | $F_{1,14}$ = 58.39 | <0.001 |
| 4E | $F_{1,14}$ = 88.28 | <0.001 | $F_{1,14}$ = 141.77 | <0.001 | $F_{1,14}$ = 192.28 | <0.001 | $F_{1,14}$ = 13.81 | 0.002 | $F_{1,14}$ = 92.88 | <0.001 |
| 4F | $F_{1,14}$ = 19.13 | <0.001 | $F_{1,14}$ = 47.93 | <0.001 | $F_{1,14}$ = 31.30 | <0.001 | $F_{1,14}$ = 0.057 | 0.815 | $F_{1,14}$ = 17.46 | <0.001 |
| 4G | $F_{1,14}$ = 85.01 | <0.001 | $F_{1,14}$ = 117.32 | <0.001 | $F_{1,14}$ = 194.74 | <0.001 | $F_{1,14}$ = 4.16 | 0.061 | $F_{1,14}$ = 89.44 | <0.001 |

control difficulty. The coefficient of determination ($l^2$) between these two variables was calculated according to following formula for calculating Pearson's correlation coefficients.

$$l^2 = S_{dxdy}^2/(S_{dxdx} \times S_{dydy})$$

$S_{dxdy}$, sum of products
$S_{dxdx}$ and $S_{dydy}$, sum of squares
Null hypothesis, correlation coefficient = 0
Alternative hypothesis, correlation coefficient ≠ 0
Test statistic ($t$) = $l/\sqrt{1 - l^2}/(n - 1)$
Degree of freedom = n − 1
Differences were considered significant when a $|t|$ value was larger than 1.697.

Changes in PSD of EEG which was significantly correlated with the increase in amplitude of high-frequency bandwidth across postural control task 1 and 2, were considered to be common cortical activities related to postural maintenance *via* the sensory feedback mechanism.

# RESULTS

## The duration of eye closure and change in stance after the sound indications

We established a system in which the displacement of COP, EEG, loading pressure of feet and the motion of gaze point were simultaneously measured under the electronically shielded and shaded space (Fig. 1A). To directly compare the effects of stance on cortical activity, the stance was continuously changed from stable (BP standing) to unstable (UP standing) under EO and EC conditions (Fig. 1B). In addition, the participants also performed other postural control tasks in which the standing pattern was continuously changed from stable (EO condition) to unstable (EC condition) under BP and UP conditions (Fig. 1C). We aimed to explore the commonality of the relationship between postural stability and EEG using these two postural control tasks.

To verify the loading pressure on the sole during UP standing and eye closing, we first examined the change in loading pressure on the sole and gaze point. The COP displacement was remarkably changed in the ML direction, but not in the AP direction (Fig. $2A_1$). The loading pressure on the sole was rapidly decreased by the change in stance and was maintained at almost zero, suggesting that dominant foot was sustainably raised during UP standing (Fig. $2A_2$). To investigate individual differences in the latency from sound indication to start of postural change, 4 s epoch (29–33 s) was extracted from time-series data of loading pressure on dominant foot, and two breaking points were statistically detected. The duration between sound indication and $BK_1$, when the loading pressure on dominant foot began to drop, was small and there was no significant difference between EO and EC conditions (Fig. 2B; $t_{14} = 2.030$, $P = 0.062$). The duration between $BK_1$ and $BK_2$, when the loading pressure on the dominant foot became almost zero, was not significant difference between EO and EO conditions (Fig. 2B; $t_{14} = 1.04$, $P = 0.315$). To investigate the latency from sound indication to eye closure, we examined the time

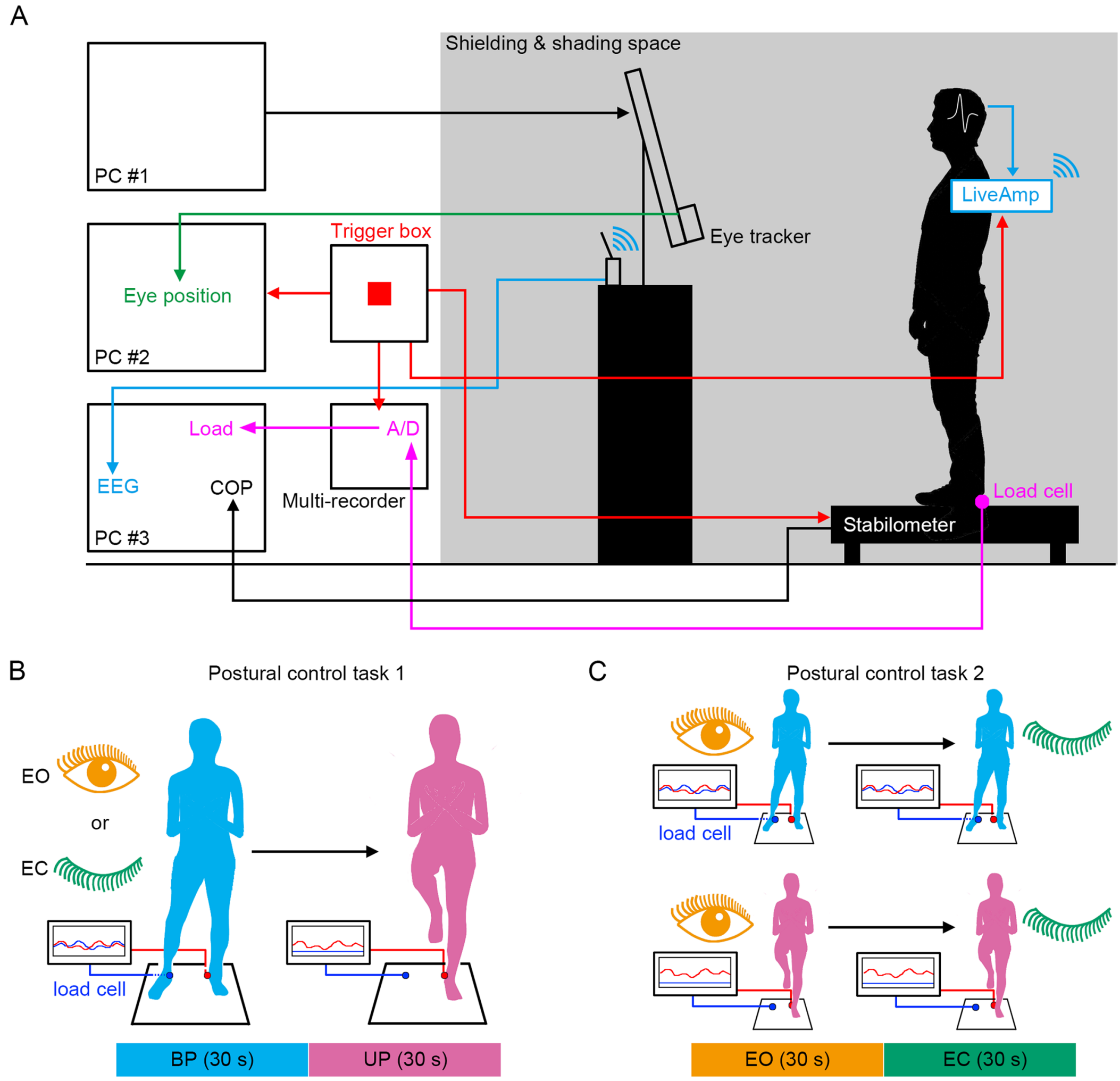

**Figure 1 Experimental procedures.** (A) The simultaneous measurement system of the displacement of center of pressure (COP), load of pressure, electroencephalogram (EEG), and eye position. (B) Schematic representation of postural control task 1. (C) Schematic representation of postural control task 2.

point of eye closure observed in the temporal coordinates of gaze point during BP and UP standing (*Wei et al., 2017*). The movement of gaze point was completely stopped by eye closure along x and y directions on the monitor during both BP and UP standing, because the center of the pupil and the reflection of the illuminator on the cornea could not be observed by eye closure (Figs. $2C_1$ and $2C_2$). Interestingly, the duration between sound indication and stopping of gaze point was significantly shorter during UP standing than during BP standing (Fig. 2D; $t_{14}$ = 2.444, $P$ = 0.028). These results indicate that all participants adjust their posture with standing on one leg or with eye closed after sound indications. In addition, the data of 2 s after the sound indication includes non-stationary responses caused by the postural transition. To focus on stationary responses in this study, the data of 2 s before and after the sound indication was eliminated.

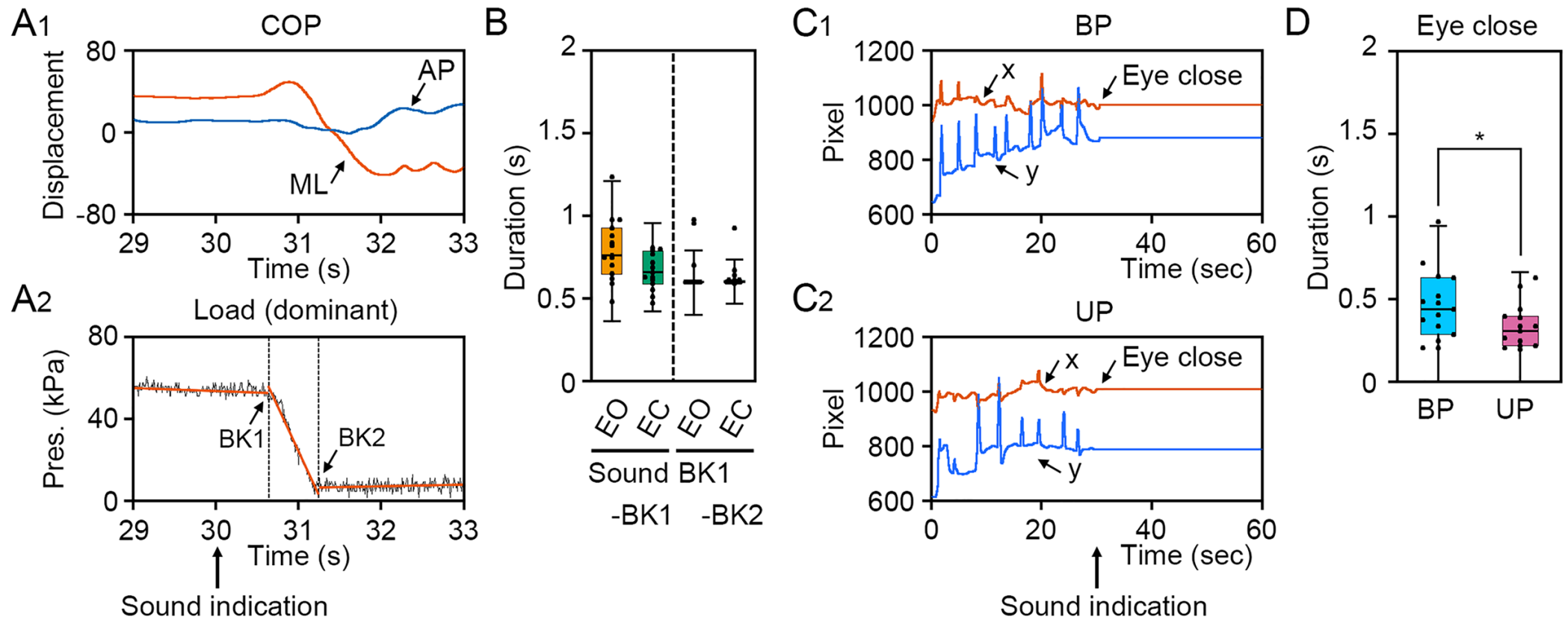

**Figure 2 Identification of time point of postural transition and eye closure.** ($A_1$) Representative change in the center of pressure (COP) displacement in the anteroposterior (AP, blue) and mediolateral (ML, red) directions during bipedal (BP, cyan background) and unipedal (UP, magenta background) standing. ($A_2$) Representative change in load of pressure on dominant (right) foot. Breaking points (BKs) were determined using a strucchange package in R. Black represents the load of pressure on dominant foot. Red represents linear regression line. (B) Time point of BK1 and BK2 under eye open (EO, orange) and eye closed (EC, green) conditions. (C) Variation of BK values under EO and EC conditions. (D) Representative temporal coordinates of gaze point in x (red) and y (blue) axes under EO (orange background) and EC (green background) conditions. (E) Time point of eye closure during BP and UP standing. (F) Variation of eye closing point during BP and UP standing. Box plots represent the median, first and third quartiles (boxes), and 1.5 × interquartile ranges (IQRs) (whiskers). Number of participants: $n = 15$. Statistical differences were analyzed using Paired *t*-test. Abbreviations: AP, anteroposterior; BK, breaking point; BP, bipedal; EC, eye closed; EO, eye open; ML, mediolateral; Pres, pressure; UP, unipedal. Statistical significance is indicated by asterisk: $^*P < 0.05$.

## Effects of change in standing patterns on postural control

Next, we examined the effects of change in standing patterns on COP displacement under both EO and EC conditions. COP displacement was small during the BP standing period, but immediately increased when UP standing under the EO condition (Fig. 3A). Under the EC condition, the COP displacement during the BP standing period was similar to that under the EO condition. However, the COP displacement during UP standing was greater under the EC condition than under the EO condition (Fig. 3B). Based on results of loading data processing, the COP data of 2 s before and after the sound indication including non-stationary responses was eliminated, and COP data including remaining 28 s were used for statistical analysis. Detailed statistical data was shown in Table 1. There was significant interaction between two independent variables in total path length, AP path length, ML path length, AP amplitude, and ML amplitude ($P < 0.001$). In following statistical analysis of simple main effects, the total path length of COP during UP standing was larger than that during BP standing under both EO and EC conditions (Fig. 3C; $P < 0.001$). The total path length of COP was also increased by eye closure during both BP and UP standing ($P < 0.001$). The path length of COP in the AP direction during UP standing was larger than that during BP standing under both EO and EC conditions (Fig. 3D; $P < 0.001$). The path length of COP in the AP direction was significantly increased by eye closure during both BP and UP standing ($P < 0.001$). The path length of COP in the ML direction during UP standing was larger than that during BP standing

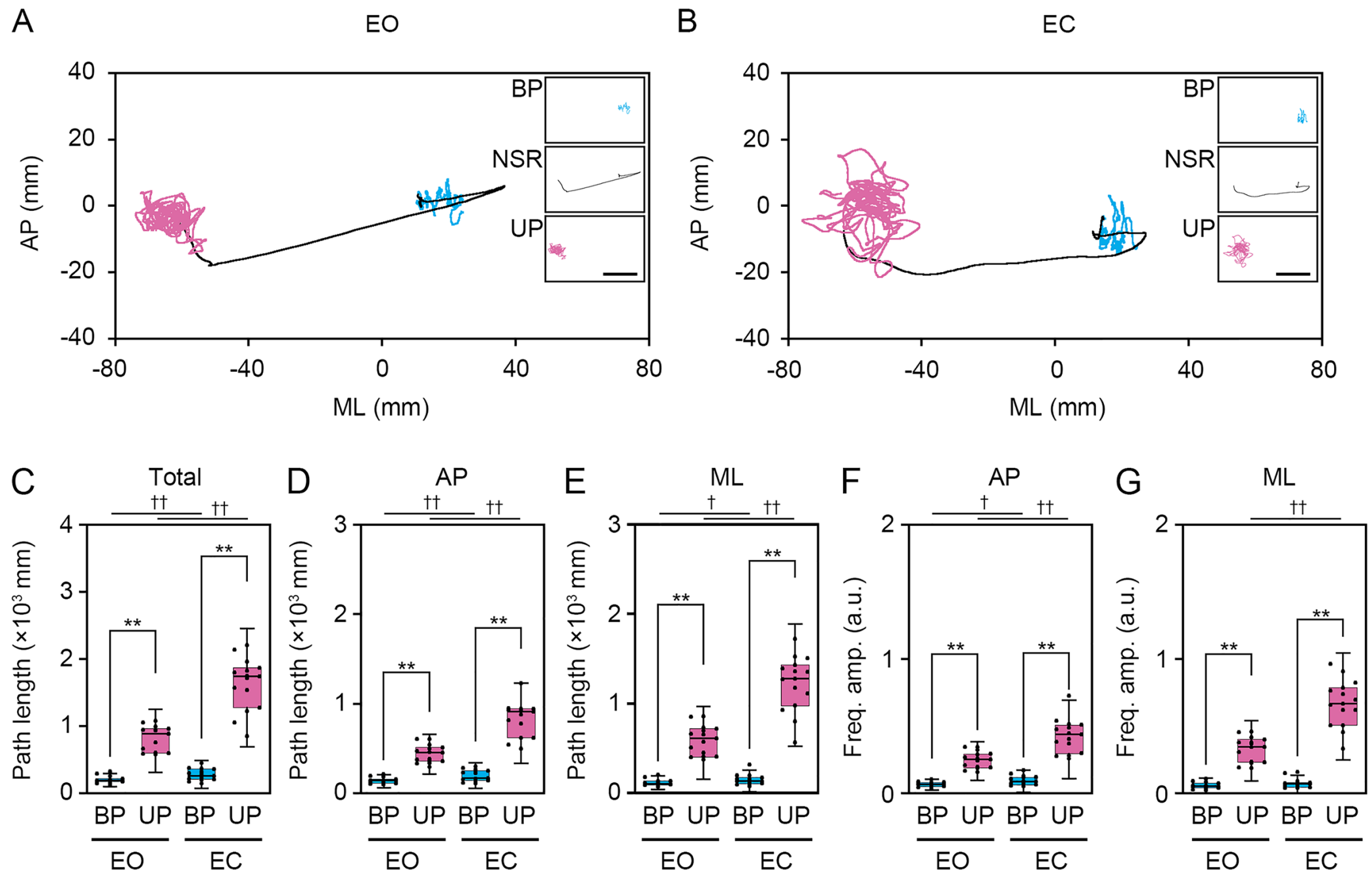

**Figure 3 Change in standing patterns affected the path length and the amplitude of the center of pressure displacement in high-frequency bandwidth.** (A, B) Representative trajectory of the center of pressure (COP) during bipedal (BP, cyan) and unipedal (UP, magenta) standing periods under the eye open (EO, A) and eye closed (EC, B) conditions. (C) Total path lengths of COP during BP and UP standing periods under the EO and EC conditions. (D) Path lengths of COP in the anteroposterior (AP) direction during BP and UP standing periods under the EO and EC conditions. (E) Path lengths of COP in the mediolateral (ML) direction during BP and UP standing periods under the EO and EC conditions. (F) Alterations in the amplitudes of the COP displacement in high-frequency bandwidth (Freq.amp.) in the AP direction during BP and UP standing periods under the EO and EC conditions. (G) Alterations in the amplitudes of the COP displacement in high-frequency bandwidth (Freq.amp.) in the ML direction during BP and UP standing periods under the EO and EC conditions. Box plots represent the median, first and third quartiles (boxes), and 1.5 × interquartile ranges (IQRs) (whiskers). Number of participants: $n$ = 15. Statistical differences were analyzed using two-way ANOVA with *post-hoc* Bonferroni test. Abbreviations: AP, anteroposterior; BP, bipedal; EC, eye closed; EO, eye open; ML, mediolateral; NSR, non-stationary response; UP, unipedal. Statistical significance: $^{**}P < 0.01$ (*vs.* BP), $^{\dagger}P < 0.05$ (*vs.* EO), $^{\dagger\dagger}P < 0.01$ (*vs.* EO).

under both EO and EC conditions (Fig. 3E; $P < 0.001$). The path length of COP in the ML direction was significantly increased by eye closure during both BP ($P = 0.02$) and UP ($P < 0.001$) standing. We next examined the change in amplitude of high-frequency bandwidth of the COP displacement in AP and ML directions. The amplitude of high-frequency bandwidth in the AP direction during UP standing was higher than that during BP standing under both EO and EC conditions (Fig. 3F; $P < 0.001$). The amplitude of high-frequency bandwidth in the AP direction was significantly altered by eye closure during BP standing ($P = 0.015$), while it was remarkably increased during UP standing ($P < 0.001$). By contrast, the amplitude of high-frequency bandwidth in the ML direction during UP standing was higher than that during BP standing under both EO and EC conditions (Fig. 3G; $P < 0.001$). The amplitude of high-frequency bandwidth in the ML direction was not altered by eye closure during BP standing ($P = 0.101$), while it was

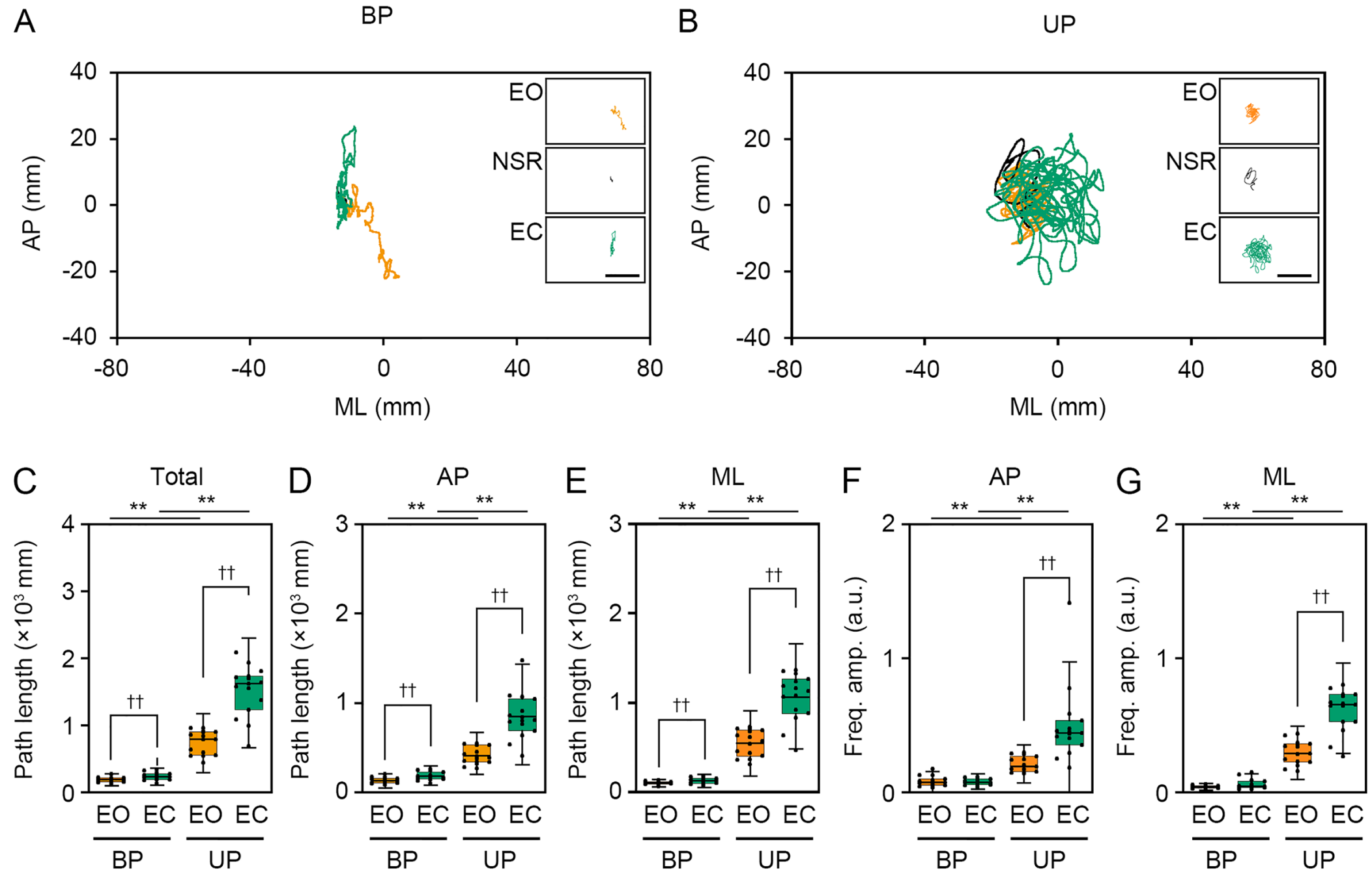

**Figure 4 Visual deprivation affected the path length and the amplitude of the center of pressure displacement in high-frequency bandwidth.** (A, B) Representative trajectory of the center of pressure (COP) during eye open (EO, orange) and eye closed (EC, green) periods under bipedal (BP, A) and unipedal (UP, B) standing conditions. (C) Total path lengths of COP during EO and EC periods under BP and UP standing conditions. (D) Path lengths of COP in the anteroposterior (AP) direction during EO and EC periods under BP and UP conditions. (E) Path lengths of COP in the mediolateral (ML) direction during EP and EC periods under BP and UP conditions. (F) Alterations in the amplitudes of the COP displacement in high-frequency bandwidth (Freq.amp.) in the AP direction during EO and EC periods under BP and UP standing conditions. (G) Alterations in the amplitudes of the COP displacement in high-frequency bandwidth (Freq.amp.) in the ML direction during EO and EC periods under BP and UP standing conditions. Box plots represent the median, first and third quartiles (boxes), and 1.5 × interquartile ranges (IQRs) (whiskers). Number of participants: $n = 15$. Statistical differences were analyzed using repeated two-way ANOVA with *post-hoc* Bonferroni test. Abbreviations: AP, anteroposterior; BP, bipedal; EC, eye closed; EO, eye open; ML, mediolateral; NSR, non-stationary response; UP, unipedal. Statistical significance: $^{*}P < 0.05$ (vs. BP), $^{**}P < 0.01$ (*vs.* BP), $^{\dagger\dagger}P < 0.01$ (*vs.* EO).

significantly increased during UP standing ($P < 0.001$). Taken together, these data indicate that the path length of COP and the amplitude of high-frequency bandwidth of COP displacement were increased by the change in stance under both EO and EC conditions.

## Effects of visual deprivation on postural control

We examined the effects of visual deprivation on the COP displacement during both BP and UP standing. The COP displacement was small under the EO condition during BP standing, and it was almost unchanged under the EC condition (Fig. 4A). During UP standing, the COP displacement was larger than that during BP standing, and this was additionally enlarged by eye closure (Fig. 4B). Similar to postural control task 1, the COP data of 2 s before and after the sound indication was eliminated, and COP data of remaining 28 s were used for statistical analysis. Detailed statistical data was shown in

Table 1. There was significant interaction between two independent variables in total path length, AP path length, ML path length, AP amplitude, and ML amplitude ($P < 0.001$). In following statistical analysis of simple main effects, the total path length of COP was significantly increased by eye closure during both BP and UP standing (Fig. 4C; $P < 0.001$). Similar to in postural control task 1, the total path length of COP during UP standing was significantly larger than that during BP standing under both EO and EC conditions ($P < 0.001$). The path length of COP in the AP direction was also increased by eye closure during both BP and UP standing (Fig. 4D; $P < 0.001$). Similar to in postural control task 1, the path length of COP in the AP direction during UP standing was significantly larger than that during BP standing under both EO and EC conditions ($P < 0.001$). The path length of COP in the ML direction was also significantly increased by eye closure during both BP and UP standing (Fig. 4E; $P < 0.001$). Similar to postural control task 1, the path length of COP in the ML direction during UP standing was significantly larger than that during BP standing under both EO and EC conditions ($P < 0.001$). We next examined the change in amplitude of high-frequency bandwidth of the COP displacement in AP and ML directions. The amplitude of high-frequency bandwidth in the AP direction was not altered by eye closure during BP standing ($P = 0.815$), but it was significantly increased during UP standing (Fig. 4F; $P < 0.001$). The amplitude of high-frequency bandwidth of the COP displacement in the AP direction was significantly larger during UP standing than during BP standing under both EO and EC condition ($P < 0.001$). The amplitude of high-frequency bandwidth in the ML direction was also not altered by eye closure during BP standing ($P = 0.061$), but it was significantly increased during UP standing (Fig. 4G; $P < 0.001$). The amplitude of high-frequency bandwidth of the COP displacement in the ML direction during UP standing was larger than that during BP standing under both EO and EC condition ($P < 0.001$). Taken together, these data indicate that the path length of COP and the amplitude of high-frequency bandwidth of COP displacement were increased by eye closure during UP standing.

## Change in PSD of EEG during postural control task 1

Cortical activity has been shown to be closely related to postural control (*St George et al., 2021*; *Sugihara et al., 2021*). Therefore, we next examined the change in EEG at cortical 8 position. The common changes in cortical activities between postural control task 1 and 2 were representatively shown in main figures. Detailed statistical data was shown in Tables 2–5. First, we examined the effects of standing pattern and eye closure on the PSD of theta band activity (Figs. 5A and S1). There was no interaction between two independent variables (stance and eye) in Fp1 ($P = 0.06$), F3 ($P = 0.223$), C3 ($P = 0.816$), C4 ($P = 0.78$), and P3 ($P = 0.051$) channels (Table 3). The PSD of theta band was significantly higher during UP standing than during BP standing in Fp1 ($P < 0.001$), F3 ($P < 0.001$), C3 ($P = 0.004$), and C4 ($P = 0.007$) channels. The PSD of theta band was significantly higher under the EC condition than under the EO condition in C3 ($P = 0.005$), C4 ($P = 0.004$), and P3 ($P = 0.033$) channels. On the other hand, there were significant interactions in Fp2 ($P = 0.008$), F4 ($P = 0.029$), and P4 ($P = 0.048$) channels (Table 5). In following statistical analysis of simple main effects, the PSD of theta band was significantly increased by change

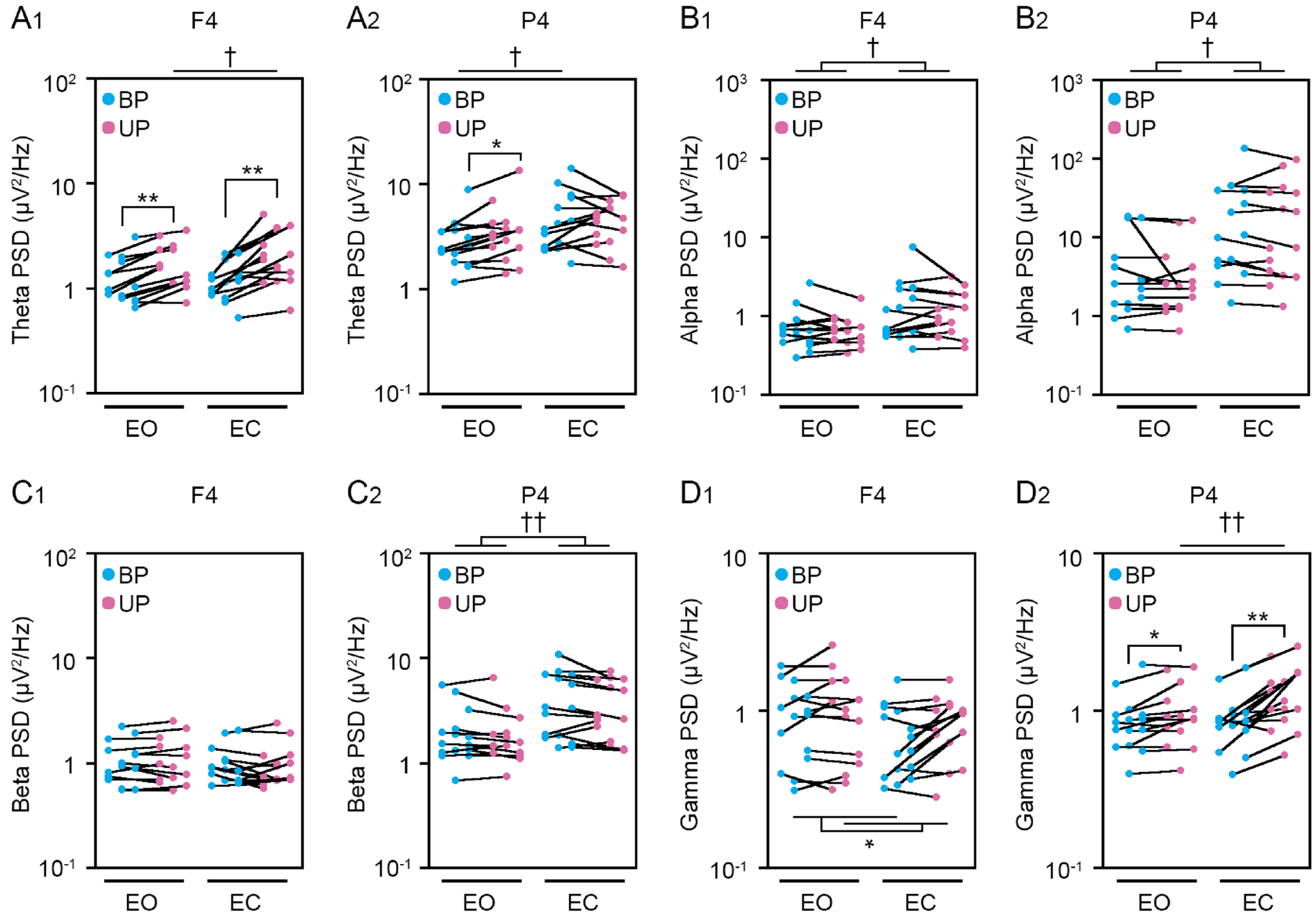

**Figure 5 Change in standing patterns affected the power spectral densities of electroencephalograms.** (A) Alterations in the power spectral densities (PSDs) of theta band in F4 ($A_1$) and P4 ($A_2$) channels during BP (cyan) and UP (magenta) standing periods under EO and EC conditions. (B) Alterations in the PSDs of alpha band in F4 ($B_1$) and P4 ($B_2$) channels during BP (cyan) and UP (magenta) standing periods under EO and EC conditions. (C) Alterations in the PSDs of beta band in F4 ($C_1$) and P4 ($C_2$) channels during BP (cyan) and UP (magenta) standing periods under EO and EC conditions. (D) Alterations in the PSDs of gamma band in F4 ($D_1$) and P4 ($D_2$) channels during BP (cyan) and UP (magenta) standing periods under EO and EC conditions. Plots represent the data obtained from individual participants. Number of participants: $n$ = 15. Statistical differences were analyzed using repeated two-way ANOVA test. Abbreviations: AP, anteroposterior; BP, bipedal; EC, eye closed; EO, eye open; PSD, power spectral density; UP, unipedal. Statistical significance: $^{*}P < 0.05$ (*vs*. BP), $^{**}P < 0.01$ (*vs*. BP), $^{\dagger}P < 0.05$ (*vs*. EO), $^{\dagger\dagger}P < 0.01$ (*vs*. EO).

in stance in Fp2 (EO, $P = 0.001$; EC, $P < 0.001$), F4 (EO, $P < 0.001$; EC, $P = 0.002$), and P4 (EO, $P = 0.016$) channels. The PSD of theta band was significantly higher under the EC condition than under the EO condition in Fp2 (UP, $P = 0.002$), F4 (UP, $P = 0.012$), and P4 (BP, $P = 0.039$). Second, the PSD of alpha band was compared between BP and UP standing under EO and EC conditions (Figs. 5B and S1). There was also no interaction between two independent variables in all channels (Tables 2 and 3). The PSD of alpha band was significantly lower during UP standing than BP standing in C3 ($P = 0.043$) channel. By contrast, the PSD of alpha band was significantly higher under the EC condition than under the EO condition in F3 ($P = 0.025$), F4 ($P = 0.04$), C3 ($P = 0.007$), C4 ($P = 0.007$), P3 ($P = 0.019$), and P4 ($P = 0.018$) channels (Tables 2 and 3). Third, the PSD of beta band was compared between BP and UP standing under EO and EC condition (Figs. 5C and S1). There was no interaction between two independent variables in all channels (Tables 2 and 3). Standing pattern did not affect the PSD of beta band in all channels. By contrast, the PSD of beta band was significantly higher under the EC condition than under the EO

condition in C3 ($F_{1,14}$ = $P < 0.001$), C4 ($P = 0.002$), P3 ($P < 0.001$), and P4 ($P = 0.002$) channels (Tables 2 and 3). Fourth, the PSD of gamma band was compared between BP and UP standing under EO and EC conditions (Figs. 5D and S1). There was also no interaction between two independent variables in Fp1 ($P = 0.626$), Fp2 ($P = 0.368$), F3 ($P = 0.095$), F4 ($P = 0.620$), C3 ($P = 0.963$), and C4 ($P = 0.321$) channels (Tables 2 and 3). The PSD of gamma band was significantly higher during UP standing than during BP standing in Fp1 ($P = 0.008$), F3 ($P = 0.011$), F4 ($P = 0.013$), and C4 ($P = 0.019$) channels (Tables 2 and 3). On the other hand, there were significant interactions in P3 ($P = 0.024$) and P4 ($P = 0.004$) channels (Table 5). In following statistical analysis of simple main effects, the PSD of gamma band was significantly increased by change in stance in P3 (EC, $P = 0.002$) and P4 (EO, $P = 0.019$; EC, $P < 0.001$) channels. In addition, the PSD of gamma band was higher under the EC condition than under the EO condition in P4 channel during UP standing ($P = 0.004$).

## Change in PSD of EEG during postural control task 2

We next examined the effects of eye closure on the PSDs of EEG at eight cortical position during BP and UP standing. Detailed statistical data was also shown in Tables 2 and 4. First, the PSD of theta band was compared between EO and EC conditions during BP and UP standing (Figs. 6A and S2). There was no interaction between two independent variables in all channels (Tables 2 and 4). The PSD of theta band was significantly higher under the EC condition than under the EO condition in Fp1 ($P = 0.002$), Fp2 ($P < 0.001$), F3 ($P < 0.001$), F4 ($P < 0.001$), C3 ($P < 0.001$), and C4 ($P = 0.004$) channels (Tables 2 and 4). In addition, the PSD of theta band was significantly higher during UP standing than during BP standing in Fp1 ($P = 0.03$), Fp2 ($P = 0.017$), and F4 ($P = 0.005$) channels (Tables 2 and 4). Second, the PSD of alpha band was compared between EO and EC conditions during BP and UP standing (Figs. 6B and S2). There was also no interaction between two independent variables in all channels (Tables 2 and 4). The PSD of alpha band was significantly higher under the EC condition than under the EO condition in F3 ($P = 0.021$), F4 ($P = 0.02$), C3 ($P = 0.011$), C4 ($P = 0.006$), P3 ($P = 0.031$), and P4 ($P = 0.021$) (Tables 2 and 4). In addition, standing pattern did not affect the PSD of alpha band in all channels (Tables 2 and 4). Third, the PSD of beta band was compared between EO and EC conditions during BP and UP standing (Figs. 6C and S2). There was no interaction between two independent variables in all channels (Tables 2 and 4). The PSD of beta band was significantly higher under the EC condition than under the EO condition in Fp1 ($P = 0.029$), F3 ($P = 0.004$), C3 ($P = 0.007$), C4 ($P = 0.003$), P3 ($P = 0.001$), and P4 ($P = 0.004$) channels (Tables 2 and 4). In addition, standing pattern did not affect the PSD of beta band in all channels, and these were also common to postural control task 1 (Tables 2 and 4). Fourth, the PSD of gamma band was compared between EO and EC conditions during BP and UP standing (Figs. 6D and S2). There was no interaction between two independent variables in Fp1 ($P = 0.507$), Fp2 ($P = 0.643$), F3 ($P = 0.146$), F4 ($P = 0.449$), and C3 ($P = 0.120$) channels (Tables 2 and 4). The PSD of gamma band was not altered by eye closure in these channels (Tables 2 and 4). By contrast, the PSD of gamma band was significantly higher during UP standing than during BP standing in Fp1

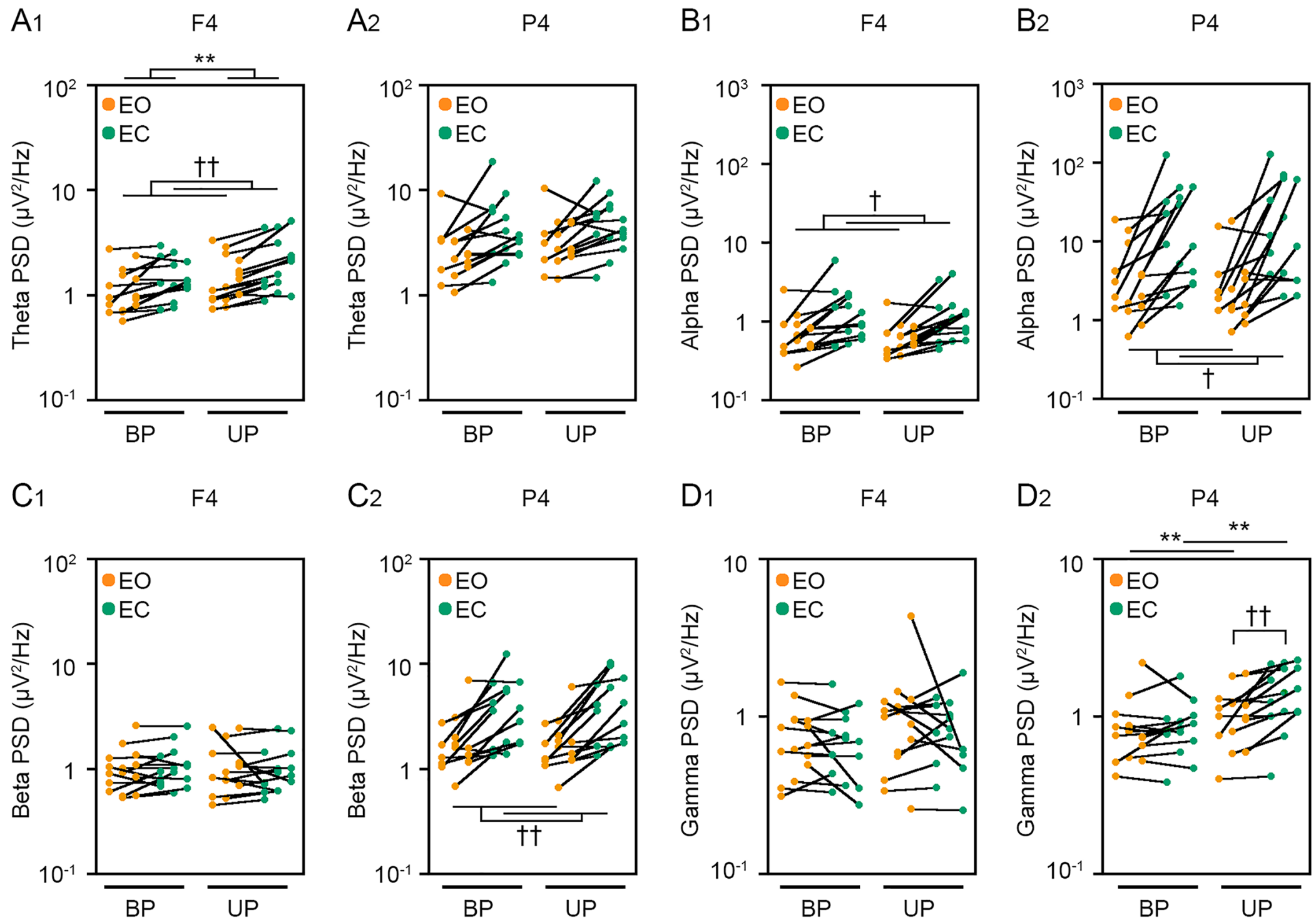

**Figure 6 Visual deprivation affected the power spectral densities of electroencephalograms.** (A) Alterations in the power spectral densities (PSDs) of theta band in F4 ($A_1$) and P4 ($A_2$) channels during EO (orange) and EC (green) periods under BP and UP standing conditions. (B) Alterations in the PSDs of alpha band in F4 ($B_1$) and P4 ($B_2$) channels during EO (orange) and EC (green) periods under BP and UP standing conditions. (C) Alterations in the PSDs of beta band in F4 ($C_1$) and P4 ($C_2$) channels during EO (orange) and EC (green) periods under BP and UP standing conditions. (D) Alterations in the PSDs and pairwise distributions of gamma band in F4 ($D_1$) and P4 ($D_2$) channels during EO (orange) and EC (green) periods under BP and UP standing conditions. Plots represent the data obtained from individual participants. Number of participants: $n = 15$. Statistical differences were analyzed using repeated two-way ANOVA test. Abbreviations: AP, anteroposterior; BP, bipedal; EC, eye closed; EO, eye open; PSD, power spectral density; UP, unipedal. Statistical significance: $^{**}P < 0.01$ (*vs.* BP), $^{\dagger}P < 0.05$ (*vs.* EO), $^{\dagger\dagger}P < 0.01$ (*vs.* EO).

($P = 0.022$) (Table 4). There were significant interactions in C4 ($P = 0.047$), P3 ($P = 0.030$) and P4 ($P = 0.007$) (Table 5). In following statistical analysis of simple main effects, the PSD of gamma band was significantly higher under the EC condition than under the EO condition in C4 ($P = 0.043$), P3 ($P = 0.016$) and P4 ($P = 0.002$) channels during UP standing. In addition, the PSD of gamma band was significantly higher during UP standing than during BP standing under the EO condition in P4 channel ($P = 0.007$). Moreover, the PSD of gamma band was significantly higher during UP standing than during BP standing under the EC condition in P3 ($P = 0.007$) and P4 ($P < 0.001$) channels.

## Similarity of changes in PSD of EEG during postural control task 1 and 2

To examine common changes in the cortical activity during postural control task 1 and 2, we first calculated differences in PSD of EEG before and after the sound indication (Fig. 7). In general, the PSDs of frontal theta band were increased in both postural control tasks. On the other hand, the PSDs of alpha band were decreased during postural control task 1,

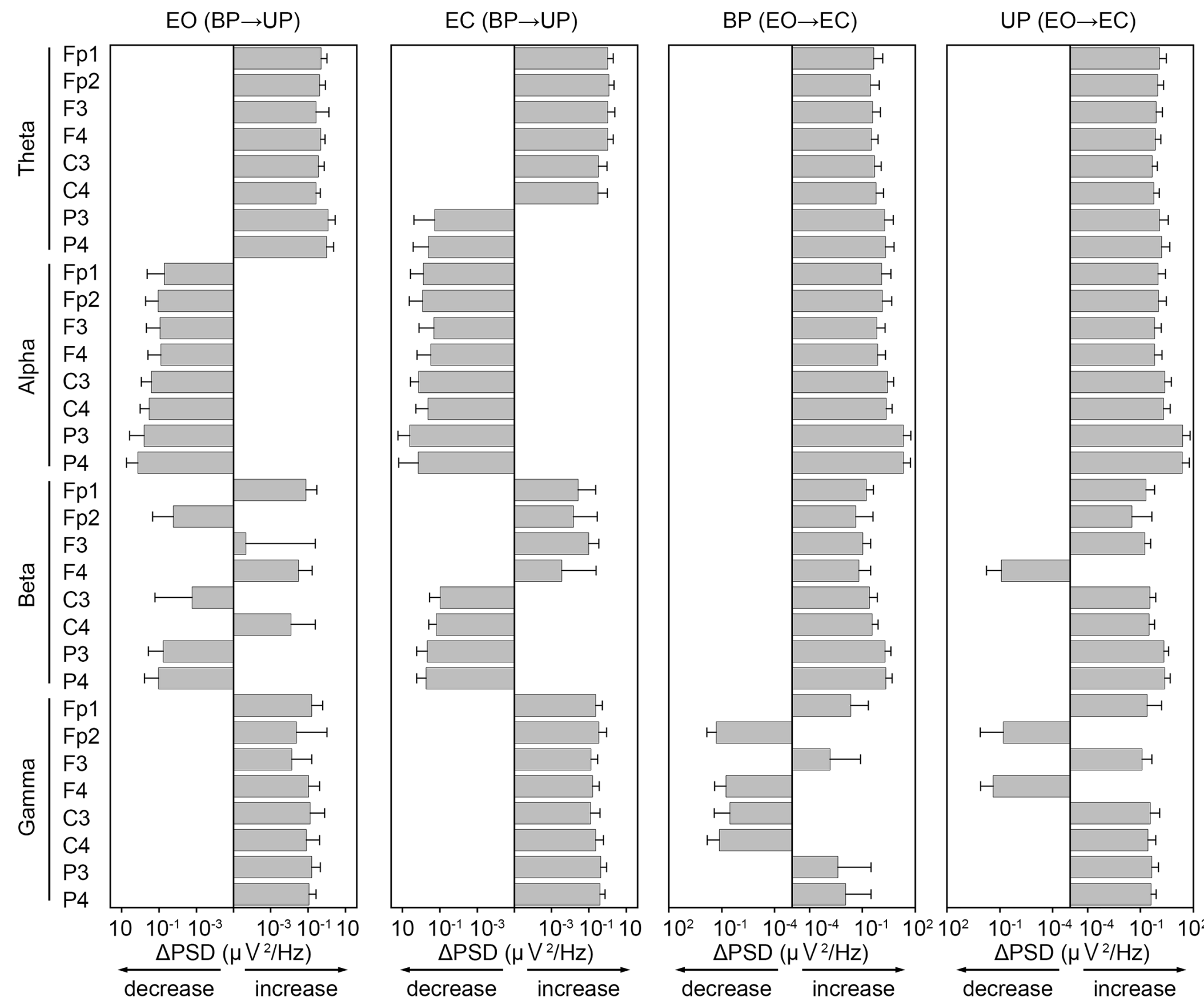

**Figure 7 Similarity of the change in power spectral density of electroencephalogram.** The differences of power spectral density (ΔPSD) of electroencephalogram in theta, alpha, beta, and gamma bandwidths at each channel before and after the sound indication. Number of participants: $n$ = 15. Error bars represent the standard deviation.

while those were increased during postural control task 2. The changes in PSD of beta band appeared to be less common between postural control task 1 and 2. The PSDs of parietal gamma band were increased in both postural control task 1 and 2.

## Correlation between the change in amplitude of high-frequency bandwidth of the COP displacement and the change in PSD of EEG

To statistically reveal common changes in EEG related to the postural maintenance during postural control task 1 (PCT1) and 2 (PCT2), we calculated coefficients of determination ($l^2$) between the change in amplitude of high-frequency bandwidth of the COP displacement in AP and ML directions and the change in PSDs of EEG. Following three experimental conditions were selected as postural control tasks that continuous maintenance of balance was required, because the high-frequency components of the COP displacement was significantly increased. Then, we explored the common correlation between the change in amplitude of high-frequency bandwidth of the COP displacement and the change in PSDs of EEG (Figs. 8 and 9).

(1) Change in standing pattern under the EO condition (PCT1 (EO))

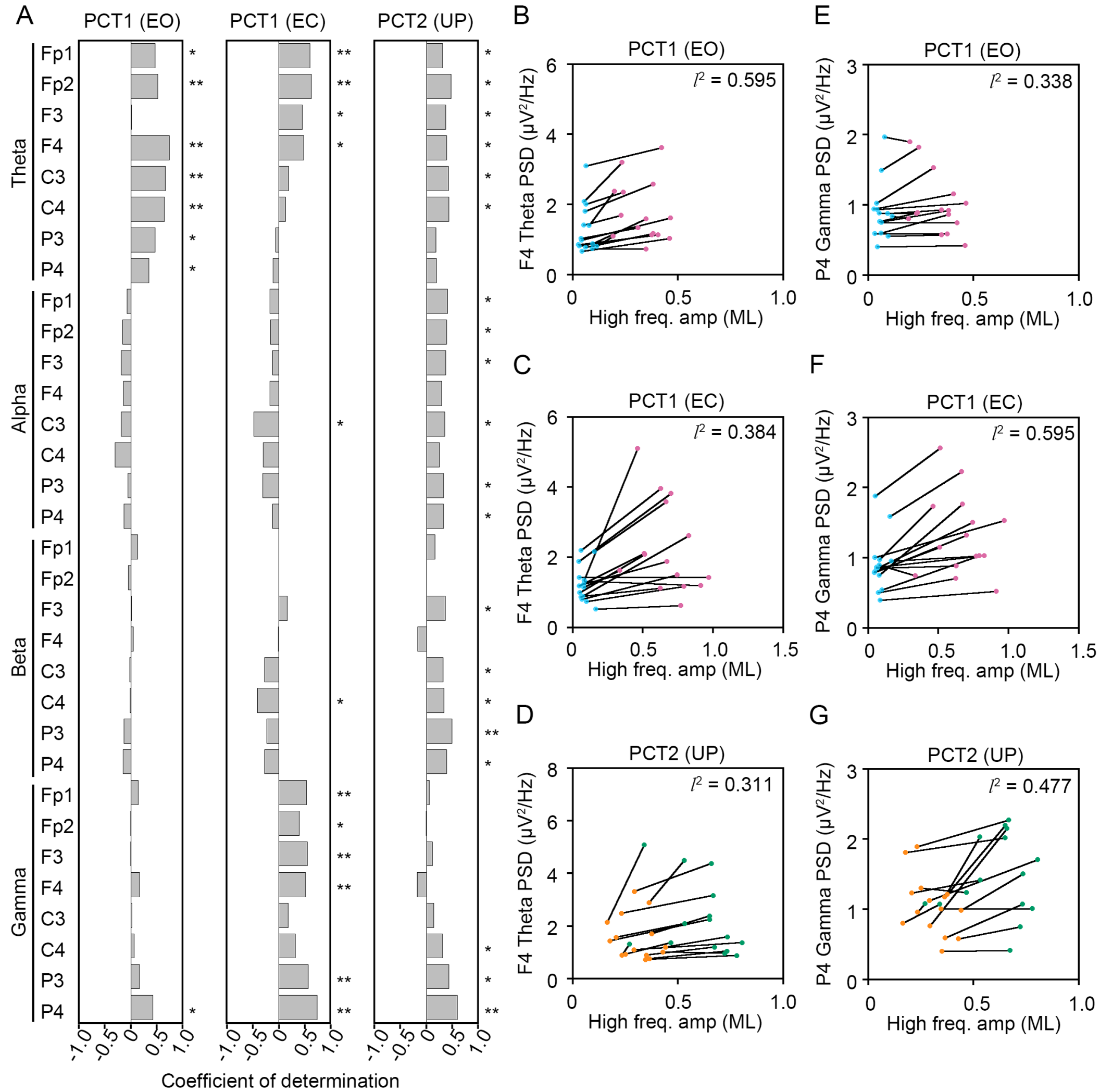

**Figure 8 Correlation between the change in amplitudes of the center of pressure displacement in the anteroposterior direction and electroencephalogram power.** (A) Coefficients of determination between the change in amplitude of the center of pressure (COP) displacement in high-frequency bandwidth in the anteroposterior (AP) direction and power spectral densities (PSDs) of electroencephalogram (EEG) in theta, alpha, beta, and gamma bandwidths in postural control task (PCT) 1 and 2 under the eye open (EO), eye closed (EC), and unipedal (UP) conditions. (B) The correlation plots between the amplitude of COP displacement in high-frequency bandwidth in the AP direction and the PSD of EEG in theta band in the frontal area (F4 channel) during PCT1 under the EO condition. (C) The correlation plots between the amplitude of COP displacement in high-frequency bandwidth in the AP direction and the PSD of EEG in theta band in F4 channel during PCT1 under the EC condition. (D) The correlation plots between the amplitude of COP displacement in high-frequency bandwidth in the AP direction and the PSD of EEG in theta band in F4 channel during PCT2 during UP standing. (E) The correlation plots between the amplitude of COP displacement in high-frequency bandwidth in the AP direction the PSD of EEG in gamma band in P4 channel during PCT1 under the EO condition. (F) The correlation plots between the amplitude of COP displacement in high-frequency bandwidth in the AP direction the PSD of EEG in gamma band in P4 channel during PCT1 under the EC condition. (G) The correlation plots between the amplitude of COP displacement in high-frequency bandwidth in the AP direction and the PSD of EEG in gamma band in P4 channel during PCT2 during UP standing. Abbreviations: AP, anteroposterior; BP, bipedal; EC, eye closed; EEG, electroencephalogram; EO, eye open; PSD, power spectral density; PCT, postural control task; UP, unipedal. Statistical significance is indicated by asterisks: $^*P < 0.05$, $^{**}P < 0.01$.

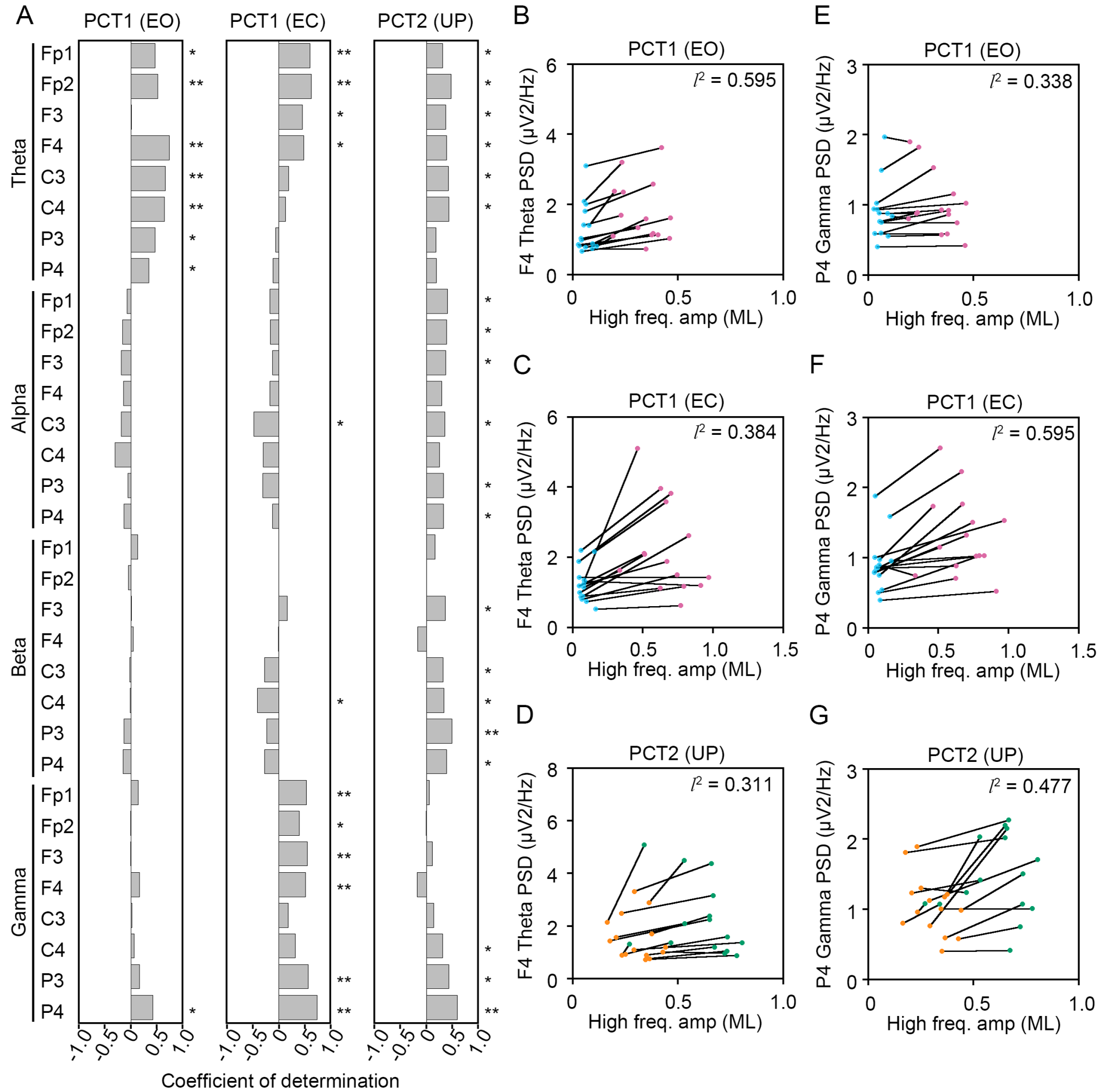

**Figure 9 Correlation between the change in amplitudes of the center of pressure displacement in the mediolateral direction and electroencephalogram power.** (A) Coefficients of determination between the change in amplitude of the center of pressure (COP) displacement in high-frequency bandwidth in the mediolateral (ML) direction and power spectral densities (PSDs) of electroencephalogram (EEG) in theta, alpha, beta, and gamma bandwidths in postural control task (PCT) 1 and 2 under the eye open (EO), eye closed (EC), and unipedal (UP) conditions. (B) The correlation plots between the amplitude of COP displacement in high-frequency bandwidth in the ML direction and the PSD of EEG in theta band in the frontal area (F4 channel) during PCT1 under the EO condition. (C) The correlation plots between the amplitude of COP displacement in high-frequency bandwidth in the ML direction and the PSD of EEG in theta band in F4 channel during PCT1 under the EC condition. (D) The correlation plots between the amplitude of COP displacement in high-frequency bandwidth in the ML direction and the PSD of EEG in theta band in F4 channel during PCT2 during UP standing. (E) The correlation plots between the amplitude of COP displacement in high-frequency bandwidth in the ML direction the PSD of EEG in gamma band in P4 channel during PCT1 under the EO condition. (F) The correlation plots between the amplitude of COP displacement in high-frequency bandwidth in the AP direction the PSD of EEG in gamma band in P4 channel during PCT1 under the EC condition. (G) The correlation plots between the amplitude of COP displacement in high-frequency bandwidth in the ML direction and the PSD of EEG in gamma band in P4 channel during PCT2 during UP standing. Abbreviations: BP, bipedal; EC, eye closed; EEG, electroencephalogram; EO, eye open; ML, mediolateral; PSD, power spectral density; PCT, postural control task; UP, unipedal. Statistical significance is indicated by asterisks: $^*P < 0.05$, $^{**}P < 0.01$.

(2) Change in standing pattern under the EC condition (PCT1 (EC))

(3) Change in visual input during UP standing (PCT2 (UP))

Significant positive correlation between the change in PSD of theta band and the change in amplitude of high-frequency bandwidth of the COP displacement in the AP direction was commonly observed across three conditions in frontal areas (Fig. 8A). For instance, the PSD of theta band in F4 channel was increased in a positive correlation with amplitude of high-frequency bandwidth of the COP displacement by the change in standing pattern across three types of postural control task described above (Figs. 8B–8D). Significant positive correlation between the change in PSD of theta band and the change in amplitude of high-frequency bandwidth of the COP displacement in the ML direction was also commonly observed across three conditions in frontal and parietal areas (Figs. 9A–9D). Significant positive correlation between the change in PSD of parietal gamma band and the change in amplitude of high-frequency bandwidth of the COP displacement in the AP direction was commonly observed across three conditions (Figs. 8A, 8E–8G). In addition, significant positive correlation between the change in PSD of parietal gamma band and the change in amplitude of high-frequency bandwidth of the COP displacement in the ML direction was also commonly observed across three conditions (Figs. 9A, 9E–9G).

# DISCUSSION

## Effects of increase in postural control task difficulty on COP data

The present study showed that the path length of COP displacement was significantly extended by the increase of postural control task difficulty, such as the change in stance from BP to UP and eye closure. Similar to these results, it is previously reported that the area of stabilogram is larger during UP standing than during BP standing under both EO and EC condition (*Liang, Hiley & Kanosue, 2019*). In addition, eye closure significantly affects the path length of COP displacement not only during UP standing but also during BP standing (*Asseman, Caron & Crémieux, 2005*). By contrast, the present comparative study revealed that the change in path length of COP was larger during difficult postural control tasks than during easy ones. Therefore, the strong maintenance of posture may be required during difficult postural control tasks compared with easy ones. Among the two components including in the COP motion, trembling associated with the maintenance of balance *via* the stiffness of the calf muscle and the ankle joint, and it is evaluated using high-frequency fluctuations in COP movement. We newly revealed that the amplitude of COP displacement at high-frequency bandwidth was significantly increased by the change in standing patterns and eye closure during UP standing. However, it was surprised that the amplitude of COP displacement at high-frequency bandwidth was not altered by eye closure during BP standing. There are several possible reasons for this outcome.

One possibility is that the change in postural control task difficulty caused by eye closure during BP standing is quite smaller than other postural control tasks. Another possibility is that the amplitude of COP displacement at high-frequency bandwidth is not affected by visual inputs. It is reported that visual inputs mainly influence low-frequency bandwidth of the COP displacement (*Oppenheim et al., 1999*; *Friedrich et al., 2008*; *Ongun et al., 2016*). Taken together, it is suggested that the change in amplitude of COP displacement at

high-frequency bandwidth reflects the postural maintenance required by the postural instability rather than the visual deprivation.

### Effect of changing stances on cortical EEG

In this study, the increase of theta activity in the fronto-central region was observed by the postural transition from BP to UP standing. The theta activity in the frontal area was also larger during UP standing than during BP standing in postural control task 2. Consistent with our results, theta activity has been previously reported to be increased in motor-related areas with increase of postural control task difficulty in healthy adolescents (*Hülsdünker et al., 2015*; *Edwards et al., 2018*; *Gebel, Lehmann & Granacher, 2020*). Theta activity was higher during tandem stance in young adults than regular stance on two legs, while it was not altered in older people (*Malcolm et al., 2021*). In addition, compared with healthy controls, theta activity in the frontal area was enhanced in patients that had undergone reconstruction of the anterior cruciate ligament, to maintain the postural stability during single-leg standing on an unstable platform (*An et al., 2022*). It is reported that the functional inhibition of fronto-parietal areas with continuous pulses at 50 Hz at a rate of 5 Hz (continuous theta burst stimulation) impaired postural control mechanisms to find a stable solution under challenging conditions (*Goel et al., 2019*). Therefore, it has been considered that frontal activity is important for postural control under unstable stances. On the other hand, it is recently reported that fronto-parietal theta stimulation using transcranial alternating current stimulation (tACS) improves the working memory performance under postural control conditions, but it effect on the postural control is not significant in healthy young adults (*Xiao et al., 2023*). In the present study, we first revealed that the increase of frontal theta activity was significantly correlated with the high-frequency components of the COP displacement. High-frequency components of the COP displacement are likely to reflect the stiffness of calf muscle and the ankle joint, rather than the voluntary regulation of balance through the intervention from the central nervous system (*Zatsiorsky & Duarte, 1999*, *2000*). Taken together, it is suggested that the increase of frontal theta activity is associated with the processing of sensory information fed back to the cortex, rather than voluntary regulation of balance, and it may affect attention and memory during unstable stances in healthy young adults.

### Effects of eye closure on cortical EEG

In postural control task 2, the increase of alpha activity in the fronto-central and parietal regions was observed by eye closure. The PSD of alpha activity in the fronto-central and parietal regions was also significantly larger under EC condition than under EO condition in postural control task 1. The increase of alpha band EEG power is classically reported by eye closure in the occipital area, and alpha activity is widely considered to reflect a state of cortical idling (*Chapman, Armington & Bragdon, 1962*). More recent studies suggest that the increase of alpha activity represents the functional inhibition of task irrelevant areas (*Jensen & Mazaheri, 2010*; *Foxe & Snyder, 2011*). By contrast, it is known that low-frequency power of EEG (3–20 Hz) including alpha band is increased by eye closure in

all lobes of the brain outside the visual cortex, suggesting that visual inputs affect an anatomically widely-distributed network (*Geller et al., 2014*; *Barry & De Blasio, 2017*).

On the other hand, mu rhythm (7–13 Hz) is as well studied with alpha activity, and not affected by eye opening/closure (*Rimbert et al., 2018*). For instance, postural task dependent attenuation of alpha band EEG power in parietal and occipital areas has been reported under the EO condition (*Lehmann et al., 2020*; *Büchel et al., 2021*). In addition, alpha power is also decreased by challenging sensory and postural tasks on unstable platforms (*Edwards et al., 2018*; *Lin, Hsieh & Chen, 2021*). Moreover, mental arithmetic calculation significantly attenuates the increase of alpha band EEG power under the EC condition (*Glass & Kwiatkowski, 1970*). Mu rhythm is suppressed by object-directed grasping and active walking (*Muthukumaraswamy, Johnson & McNair, 2004*; *Seeber et al., 2014*). In the present study, the PSD of alpha activity during UP standing was significantly lower than that during BP standing in C4 channel (Figs. 5, 6, S1 and S2). Thus, the reduction of alpha activity might be related to the suppression of mu rhythm. However, it may have been technically difficult to distinguish specific alpha activity related to postural maintenance from EEG data including strong alpha activity caused by eye closure. Therefore, further comparative experiments with other postural control tasks that do not include eye closure sessions are needed.

## Common increase of gamma activity associated with the maintenance of posture

In the present study, the increase of amplitude in high-frequency bandwidth of the COP displacement was used as an indicator of postural maintenance related to a sensory feedback mechanism. We observed the increase of amplitude in high-frequency bandwidth of the COP displacement during the three conditions: the transition from BP to UP under the EO condition, the transition from BP to UP under the EC condition, and the transition from EO to EC during UP standing (Figs. 3F, 3G, 4F and 4G). On the other hand, the significant increase of parietal gamma PSD was also commonly observed in these three conditions (Figs. 5 and 6). Synchronous stimulation in right fronto-parietal cortices with gamma band enveloped by theta band using tACS significantly increases the kinesthetic illusion in healthy young adults, suggesting that the fronto-parietal network has a causal role in body awareness (*Takeuchi et al., 2019*). Therefore, the increase of parietal gamma activity in these three conditions may be induced by the feedback of sensory inputs during the maintenance of balance, similar to the increase of frontal theta activity. On the other hand, the parietal somatosensory cortex reportedly projects to the frontal motor areas in mammals (*Pavlides, Miyashita & Asanuma, 1993*; *Petrof, Viaene & Sherman, 2015*). Actually, the tACS in the parietal area with gamma range improves the visuomotor performance and proprioception in the lower extremities (*Kamii, Kojima & Onishi, 2022*). Fronto-parietal theta-gamma interaction is important for the control of motor commands as well as working memory performance (*Jones, Johnson & Berryhill, 2020*; *Spooner & Wilson, 2022*). In addition, the high-frequency components of the COP are reportedly generated by the other mechanism involved in the central nervous system. The chattering dynamics of the COP displacement is induced by the intermitted control for the switching

of neural controller (*Bottaro et al., 2005*, *2008*). Further examinations of the coupling and oscillation between the parietal gamma and other cortical activities, and the intervention studies using transcranial stimulation are necessary, but the increase of parietal gamma activity is strongly indicated to be associated with the postural maintenance following periods of instability.

# CONCLUSIONS

In present study, we revealed several patterns of task-dependent alteration in EEG using two postural control tasks. The increase of theta activity in the frontal area and the increase of gamma activity in the parietal area, are commonly correlated with the increase in amplitude of high-frequency bandwidth of COP displacement in postural control task 1 and 2. Therefore, specific activation patterns of the neocortex are suggested to be important for the maintenance of posture and the processing of sensory feedback following periods of instability. However, it remains unclear whether these common cortical activities influence the output of motor commands. Neuronal activity can be modulated by external stimulation of specific cortical areas using transcranial alternating/direct current stimulation. It is necessary to investigate changes in muscular activities and the path length of COP by the transcranial modulation of neuronal activity. Our research might be beneficial in developing innovative interventions to reduce the fall risk of patients with postural control impairment.

## Limitation of this study

This study has several limitations. First, the sample size (15 participants) was small because this was intended as a pilot study. Sample size is thought to be important when linking brain scan data to behavior (*Marek et al., 2022*). Our results further needed to be replicated from EEG and COP data obtained from a large number of subjects. A second limitation is the selection bias of participants. The characteristics of the participants, including sex, age, ethnicity, and basic knowledge of postural control and brain science may have different effects on the change in COP movement and EEG caused by increasing difficulty of postural control tasks compared with young male adults. Therefore, similar experiments with female, non-Japanese people, and elderly should be needed for the generalization of our interpretations. In addition, we did not set any exclusion criteria based on physical functions. Muscular strength in lower limbs, joint range of motion of ankle joints, and habits of exercise may also affect the displacement of COP during unstable stances. A third limitation is the number of recording electrodes. In this study, we selected frontal and parietal areas, which are involved in motor regulation and sensory information processing. However, the changes in EEG in other cortical areas such as temporal and occipital areas also should be examined. Using high-density EEG systems, additional correlations between the amplitude of the COP displacement and the change in EEG may be observed in other cortical areas. In addition, this study is an observational but not an intervention study. Therefore, intervention studies using electrical stimulation of specific cortical areas are also necessary to directly prove that the increase of frontal theta and parietal gamma

activities are involved in postural adjustments. The use of high-density EEG system with more than 64 channels is recommended to discuss the exact source of the measured EEG (*Lai et al., 2005*). Elucidation of the exact source of EEG may provide useful information for intervention studies. A fourth limitation is that the present study could not exclude the contamination of artifacts derived from muscle activities, because the present study did not record muscle activity by surface electromyography. Therefore, the gamma activity measured in the present study may include some artifacts derived from muscular activity. To solve this problem, the simultaneous recording of muscle activity by electromyography is further needed to remove contamination from artifacts derived from muscle activity. A fifth limitation is the duration of task is short and the number of testing trials in a task to minimize the participants' distress. In order to clarify the relationship between the low-frequency component of the COP displacement and EEG, it is necessary to elongate the duration of postural tasks. Moreover, it is necessary to increase the number of trials (more than three times) to obtain more reliable average values. In this study, we used postural control tasks that increase in difficulty. Although the order of trials is randomized, it is necessary to examine the change in EEG and balance using postural control tasks that decrease in difficulty to further mitigate bias.

## ACKNOWLEDGEMENTS

We would like to thank Mses. Ayako Hisari and Naoko Takeuchi for reading and commenting on our manuscript, and thank Ph.D. Keigo Shiraiwa for setting of measurement systems. We also thank Mr. Benjamin Phillis (Wakayama Medical University) for the English language editing.

### Funding

This work was supported by Grants-in-Aid for Scientific Research (KAKENHI) from the Japan Society for the Promotion of Science (20K07738 and 23K10795 to Tomohiro Ohgomori). There was no additional external funding received for this study. The funders had no role in study design, data collection and analysis, decision to publish, or preparation of the manuscript.

### Grant Disclosures

The following grant information was disclosed by the authors:
Grants-in-Aid for Scientific Research (KAKENHI) from the Japan Society for the Promotion of Science: 20K07738 and 23K10795.

### Competing Interests

The authors declare that they have no competing interests.

### Author Contributions

- Shoma Ue performed the experiments, analyzed the data, prepared figures and/or tables, and approved the final draft.

- Kakeru Nakahama performed the experiments, analyzed the data, prepared figures and/or tables, and approved the final draft.
- Junpei Hayashi performed the experiments, analyzed the data, prepared figures and/or tables, and approved the final draft.
- Tomohiro Ohgomori conceived and designed the experiments, performed the experiments, analyzed the data, prepared figures and/or tables, authored or reviewed drafts of the article, and approved the final draft.

## Human Ethics

The following information was supplied relating to ethical approvals (*i.e.*, approving body and any reference numbers):

Osaka Kawasaki Rehabilitation University granted Ethical approval to carry out this study within its facilities (Ethical Application Reference number: OKRU-RA0023).

## Patent Disclosures

The following patent dependencies were disclosed by the authors:

US patent us7572008; publication date 2009-08-11.

## Data Availability

The raw measurements are available in the Supplemental Tables.

## Supplemental Information

Supplemental information for this article can be found online at http://dx.doi.org/10.7717/peerj.17313#supplemental-information.

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
