# Peer review of "Cortical activity associated with the maintenance of balance during unstable stances"

_PeerJ, doi:10.7717/peerj.17313_

## Round 0.1 · original submission · Major Revisions

After receiving feedback from three reviewers, we determined that the manuscript required significant additions and revisions before it could be accepted for publication, and we have assigned it a Major Revisoin. Please take all reviewers' comments seriously and address them promptly.

Here's my comments;
Please provide a more comprehensive discussion of limitations, especially to address potential sources of bias or confounding variables that may have influenced the results, and discuss alternative interpretations of the findings and their implications for the study's conclusions.

Reviewer 1 ·

Basic reporting

This study examines changes in cortical activity related to postural control. The present study clarified the cortical activity associated with changes in the difficulty level of a postural control task by setting two difficulty levels: from bipedal standing to unipedal standing and from open-eye to closed-eye conditions. I think the two types of difficulty settings are novel in this study. Using load cells and eye tracks to ensure that the support leg and visual difficulty level varied also adds credibility to the results of this study. Based on the above, I consider this study novel and clinically significant. Please consider revising the manuscript in light of the following comments.

Experimental design

Materials & Methods – Postural control task 1
Line 170-172
Materials & Methods – Postural control task 2
Line 191-192
Each task was performed twice.
Was the average of each trial used in the analysis?
Please add an explanation.

Materials & Methods – Correlation analysis
Line 281-294
Since correlation analysis is a statistical analysis, would it not be more appropriate to include it in the "Statistical analysis" section?

Validity of the findings

Discussion - Effect of changing stances on cortical EEG
Line 569-575
This study was conducted on healthy young individuals. Therefore, there is a limitation in considering healthy young individuals using previous studies on patients with musculoskeletal disorders and Parkinson's disease. I suggest that you refer to previous studies on healthy young individuals.

Discussion - Effects of eye closure on cortical EEG
Line 583-585
Your study shows increased alpha activity in the frontal-central and parietal regions.
Therefore, it is unlikely that increased alpha activity in the occipital region affected it.
Please modify your discussion appropriately.

Discussion - Common increase of gamma activity associated with the ability to postural adjustments
Line 612-621
I understand that parietal γ activity is involved in postural control.
However, why does γ activity increase with increasing difficulty in both bipedal standing to unipedal standing and eye open to eye closed?
Please explain the reason.

·

Basic reporting

• The introduction provides a solid foundation for the study, but it may benefit from a more detailed discussion of previous research gaps. Including more recent studies could strengthen the context.
• Some technical terms and concepts are introduced without adequate explanation, which might confuse readers unfamiliar with the field.

Experimental design

• The experimental design is well-structured, yet the manuscript could improve by further detailing the selection criteria (e.g. exclusion criteria) for participants and any control measures to mitigate bias.
• While the statistical methods employed are generally appropriate, the manuscript should better justify the choice of specific tests and discuss the assumptions underlying these statistical models. Please mention if the normality test was performed before performing the statistical analysis and if so describe the results of the normality test.

Validity of the findings

• There is a tendency to overinterpret some of the findings without sufficient statistical support. It would be beneficial to closely align conclusions with the data presented.
• The manuscript mentions limitations, but a more thorough exploration of these, including how they might impact the results or interpretations, would be valuable.
• Suggestions for future research are made, but these could be expanded to propose specific study designs or methodologies that could address remaining questions.

Additional comments

• The reference section is comprehensive, but some references are outdated. Updating these with more current research could strengthen the manuscript's foundation.
• The manuscript generally follows a clear structure, but there are instances of inconsistent formatting that should be corrected for professionalism.
• The manuscript makes a valuable contribution to its field, yet there is room for improvement in terms of clarity, depth of analysis, and the robustness of its conclusions. Addressing the above points could significantly enhance its impact and readability. It's also important to ensure that all claims are well-supported by data, and that the discussion critically engages with the study's findings in the context of the broader research landscape.

·

Basic reporting

The manuscript is written by a clear English language. However, presentation of the results can be more concise. Namely, in the main text of the manuscript, comparisons of a number of metrics for the CoP (total, AP, ML) and channel-wise EEG powers among different task conditions were reported with F and p values lengthily, which makes capturing the entire picture of the results difficult. Moreover, comparisons shown in Figs. 5 and 6 do not necessarily well correspond to the descriptions in the main text, i.e., comparisons for some EEG channels described in the main text are not shown in Figs. 5 and 6. Thus, in addition to the detailed comparisons summarized in the tables, it would be beneficial for the reader to have additional figures (box-plot comparisons) for the comparisons shown in Figs. 5 and 6 as in Figs. 7 and 8.
Please check if statistical significance indications for the comparisons between EO and EC conditions in Fig. 6 are made correctly. Maybe some are lacked.
Use of the words of path length, total distance and distance for the COP path length are mixed up throughout the manuscript. In any case, for the length or the distance, time interval for measuring the length should always be well defined. Please make it clear whenever the distance metrics are used. By the way, the vertical axis of Fig. 2A1 is labeled as “Distance (mm).” Is this correct? I think this should be displacement.
Introduction and background are well written and well referenced. However, there is a major scientific issue that should be addressed as pointed out in my section of general comments below.

Experimental design

The current research is within the scope of the journal.
Research question is well defined and meaningful. However, the statement at lines 111-113 on the “cortical activities commonly observed across multiple postural control tasks” is not well elaborated. Please describe what this statement means more in detail. Moreover, please elaborate how this issue is addressed by the method of data analysis used in this study. Namely, it is not clear enough in the method section how the way of making the balance task difficult using BP/UP and EO/EC affects commonly on the cortical activities have been assessed.

Validity of the findings

The rationale and benefit of the current research to the postural control literature are clearly stated.
Statistically analysis has been performed carefully.

Additional comments

General comments:
The current study examined CoP displacement and EEG activity during bipedal (BP) and unipedal (UP) stance with eye open (EO) and eye closed (EC) conditions. It was shown for both AP and ML directions that the high frequency components of CoP increased as the difficulty of the postural task increased by the transition from BP to UP, which was characterized by the increase in the CoP length and the spectral powers of CoP variation in the high frequency band. Such effects induced by the EO/EC conditions were more apparent for the UP condition compared to the BP condition, i.e., the high frequency components of CoP were less affected by the EO/EC visual condition.
Task-difficulty-dependent alterations in EEG were examined for theta, alpha, beta and gamma bands, along with the changes in the CoP characteristics. Powers in the theta band at frontal and parietal area and the gamma band at parietal area increased as the difficulty of the postural tasks increased by the BP/UP transition, for both EO and EC conditions. The visual condition had a greater influence on the EEG power for EC condition. The correlation between the changes in the high frequency components of CoP and the changes in the band-wise EEG powers were assessed, which showed the large positive correlations for the theta band, no matter of how the task difficulty was modulated, particularly at frontal area of the brain. When the task difficulty was increased by the EO/EC transition during UP, the alpha band power showed a positive correlation with the changes in the high frequency components of CoP. When the task difficulty was increased by the BP/UP transition during EC, the gamma band power showed a positive correlation with the changes in the high frequency components of CoP, particularly at parietal area.
The experiments and data analysis have been performed carefully, and the results of the study are novel and interesting. Thus, the manuscript deserves publicity. However, there is a critical issue that should be addressed during revision. It is associated with the background assumption made by the authors for interpreting the results of the current study. Specifically, the authors assume that the high frequency components of the CoP displacement are closely related to the ability of postural adjustments needed to recover balance, in which they considered the high frequency components of the CoP displacement as the trembling components associated with the adjustment in balance. Although the mechanisms of how the high frequency components of the CoP displacement are generated are still controversial; namely, by the elastic force due to “apparent intrinsic stiffness” that restores the instant equilibrium point (IEP) represented by the rambling component as claimed by Zatsiorsky and Duarte (1999), or by the chattering dynamics that are induced by the intermittent control for the on/off switching of the neural feedback controller as proposed by Bottaro et al (2005, 2008). It is important to notice that the former mechanism does not require a reflex or voluntary intervention from the CNS as explained by Zatsiorsky and Duarte (1999). On the other hand, the latter requires a neural control to determine a sequence of on/off switching of the neural feedback controller.
With those background in mind, my concern is the validity of the assumption of the authors considering that the high frequency components of the CoP displacement are closely related to the ability of postural adjustments needed to recover balance. Straightforward counterexample showing that a large power of the high frequency components of the CoP displacement does not imply the enhancement of the ability of postural adjustments can be seen from the spectral analysis of CoP displacement from patients with Parkinson’s disease (e.g., Matsuda et al, 2016). In this case, and in the case of current study as well, the large power for the high frequency components of the CoP displacement represents the elevated activity of calf muscles and resultant rigidity of the ankle joint (large stiffness of the ankle joint), although the neural mechanisms of elevating the activity of the calf muscle are not the same for PD patients and the healthy people performing the UP with EC. In this way, I may not agree with the background assumption considering simply that the high frequency components of the CoP displacement are closely related to the ability of postural adjustments needed to recover balance. Namely, the high frequency components of the CoP displacement are more likely to represent the stiffness of the calf muscle and the ankle joint, rather than the ability of balance recovery. In other words, the tonic activation of the calf muscles contributes to the high frequency components of the CoP displacement, and it might not operate for the phasic postural adjustments. This thought might be consistent with the original idea of trembling components of the CoP whose mechanism does not require a reflex or voluntary intervention from the CNS. In this way, I would recommend replacing the wording of “recovery of balance” in the manuscript by “maintenance of balance.” This is because the recovery of balance sounds like a phasic control of the balance, whereas the maintenance of balance sounds like a tonic stiffness control of the balance. Note that the tonic activation of the calf muscle would increase the stiffness (elasticity) of the ankle joint, leading to the enhancement of the trembling mechanism. I would like to ask the authors to consider my concern and reconsider the interpretation of the EEG response to the BP/UP transition.

- Bottaro et al., 2005: A Bottaro, M Casadio, PG Morasso, V Sanguineti, Human movement science 24 (4), 588-615, 2005.
- Bottaro et al., 2008: A Bottaro, Y Yasutake, T Nomura, M Casadio, P Morasso, Human movement science 27 (3), 473-495, 2008.
- Matsuda et al., 2016: Matsuda, K., Suzuki, Y., Yoshikawa, N., ..., Sakoda, S., Nomura, T., Proceedings of the Annual International Conference of the IEEE Engineering in Medicine and Biology Society, EMBS, 2016, 2016-October, pp. 29–32, 7590632.

---

## Round 0.2 · accepted · Accept

Since most of the points were fixed in proper way, I guess the manuscript reached the level of publication.

Reviewer 1 ·

Basic reporting

The reviewers' comments have been appropriately revised and this paper is worthy of publication.

Experimental design

None.

Validity of the findings

None.

Additional comments

None.

·

Basic reporting

No comment

Experimental design

No comment

Validity of the findings

No comment

Additional comments

Thank you for addressing all the comments and making necessary changes to the manuscript

·

Basic reporting

The authors have addressed all issues raised in my first review in positive ways.
The manuscript has been improved a lot in terms of scientific desciption and readability. I have no more comments on this interesting study.

Experimental design

I have no additional comments.

Validity of the findings

I have no additional comments.

Additional comments

I have no additional comments.